# Scanning activity of elite football players in 11 vs. 11 match play: An eye-tracking analysis on the duration and visual information of scanning

**Karl Marius Aksum** [1]*, **Lars Brotangen**[1], **Christian Thue Bjørndal** [1], **Lukas Magnaguagno**[2], **Geir Jordet**[1]

**1** Department of Sport and Social Sciences, Norwegian School of Sport Sciences, Oslo, Norway, **2** Institute of Sport Science, University of Bern, Bern, Switzerland

* kmaksum@nih.no

## Abstract

Visual perception in football ("soccer" in the U.S.) is increasingly becoming a key area of interest for researchers and practitioners. This exploratory case study investigated a subset of visual perception, namely visual exploratory scanning. The aim of this study was to examine the scanning of four elite football midfield players in an 11 vs. 11 real-game environment using mobile eye-tracking technology. More specifically, we measured the duration and information (number of teammates and opponents) of the players' scanning behavior. The results showed that the players' scanning duration was influenced by the ball context and the action undertaken with the ball at the moment of scan initiation. Furthermore, fixations were found in only 2.3% of the scans. Additionally, the results revealed that the stop point is the most information-rich part of a scan and that the players had more opponents than teammates inside their video frame during scans. Practical applications and further research recommendations are presented.

## Introduction

Visual perception is crucial for performance across different sports [1]. More specifically, the moment (when) and location (where) of information gathering is regarded as imperative when attempting to explain athletic performance [2]. Our current knowledge of visual gaze behavior in sports, and football in particular, is primarily based on studies of eye-movement registrations in laboratory settings using eye-tracking equipment [3]. These studies have provided empirical knowledge about football players' gaze behavior through the examination of fixation durations, fixation frequencies, and fixation locations in different video-simulated tasks and viewpoints between participants of different skill levels (for a review, see [4]). For example, Roca et al. [5] found that participants in an 11 vs. 11 video scenario fixated their gaze differently when the ball was near to the viewpoint of a central defender compared to when it was far away.

**Data Availability Statement:** All relevant data are within the paper and its Supporting information files.

**Funding:** The authors received no specific funding for this work.

**Competing interests:** The authors have declared that no competing interests exist.

In their review of visual perception in football, McGuckian et al. [6] found conflicting findings related to the visual perception behaviors of players at different skill levels and concluded that existing studies did not provide any clear evidence on differences in gaze behaviors. One reason for this may be the conditions of the studies, as football players' gaze behaviors have been shown to be different in laboratory studies than in more representative in situ studies [7]. This has recently led researchers to question the representativeness of the experimental tasks commonly used in studies of expert gaze behavior in dynamic sports, such as looking at screens, and how these translate to contextual sport performance [2]. Interestingly, only 31% of eye-tracking studies in high-performance sports have been conducted in the athletes' actual performance environment [2]. Therefore, Kredel et al. [3] argue that if the goal of a study is to examine gaze behavior in real-world conditions, the researchers should compromise on experimental control in favor of ensuring ecological validity. In order to bridge this gap, a recent study by Aksum et al. [8] investigated the fixations of five elite midfield players using eye-trackers in football match play and found that the players' fixation durations were much shorter than previously reported in laboratory studies. In sum, there is an apparent need for more research to be conducted in athletes' natural environments [2].

While the aforementioned branch of empirical research has focused on eye movements, another has adopted a more naturalistic approach to visual perception in football, with a focus on visual exploratory scanning, hereby referred to as scanning. This research methodology has examined visual perception in real match play at world-class levels, such as the English Premier League (EPL) [9, 10] the European Championships [11], and the U19/U17 European Championships [12]. Inspired by the ecological approach to visual perception [13], Jordet [14] suggested that in order to obtain enough information for performative football actions, players have to move their heads to direct the face (and eyes) away from the ball towards different sources of information, an activity referred to as scanning. In the ecological psychology framework, perception and action are coupled, reciprocal, and direct, meaning that human beings rely on extensive movement in order to perceive different opportunities for action [15]. According to Gibson, "We must perceive in order to move, but we must also move in order to perceive" [13]. To explain how an individual interacts with his or her environment by exploring and exploiting opportunities for action, the concept of affordances has been suggested [15]. Affordances are individual situational opportunities for action [16]. In football, affordances present themselves in all playing phases. For instance, affordances involving interaction with the ball rely heavily on the ability of players to explore their environment visually prior to engaging with the ball [17]. Thus, an ecological approach to visual perception provides us with a rich interpretive frame for investigating contextualized accounts of visual perception and movement behavior in real-world football match play. Furthermore, it greatly informed our research design because, according to the ecological approach, perception-action couplings are context-specific and have to be studied in the performance environment that the research aims to explain [18].

Visual scanning has been analyzed in a variety of field-based settings, including competitive matches [9–12], 11 vs. 11 training matches [19, 20], micro-states of play [21], and with the use of an individual pass training machine (Footbonaut) [22]. The results of these studies suggest that scanning is a contributing factor to the football performance of both youth [21] and elite players [9]. The most robust finding to date, which was found by examining 27 English Premier League players and almost 10,000 ball possessions, is that higher scan frequency prior to receiving the ball has a small but positive effect on subsequent passing performance [10]. Furthermore, scanning has also been shown to be susceptible to training with the use of imagery intervention programs [14, 23]. Additionally, one attempt was recently made to investigate scanning in a laboratory setting by placing four screens around each participant [17]. Results

showed that higher head turn frequencies before "receiving" the ball resulted in faster decision-making when players "received" the ball [17]. However, none of the studies on scanning have attempted to investigate what football players *actually* look at when they conduct a scan. Hence, our method, using eye tracking on the pitch during match play, represents a groundbreaking alternative to the current research available on scanning. Lastly, all previous studies on scanning have either measured each scan subjectively, using match videos with somewhat low video resolution that makes it difficult to detect scans (i.e., [11]), or used inertial measurement units that capture head movement, but not in relation to the ball's position (i.e., [20]). Consequently, the present study may be the first in which the objective detection and quantification of scans is possible.

Drawing upon these two different branches of research into visual gaze behavior in football, the current study is the first to investigate the scanning of elite football players in real match play using eye-tracking technology. As such, we aimed to address the absence of field study research without restrictions [2]. In doing so, the aim of this exploratory study was to add to the knowledge of visual perception in football, particularly the duration and information of scanning behavior of elite players in different naturally occurring contexts. The study results have potential practical implications for researchers, coaches, and players alike.

## Materials and methods

### Participants

We recruited four male central midfield players, aged 17 to 23 ($M$ = 20.75 years, $SD$ = 2.87), who played for two different clubs in the Norwegian Premier League (*Eliteserien*). All players were part of the first-team squad of their respective clubs. In collaboration with the coaching staff of the respective teams, we selected players based on their position as central midfielders. This selection criterion was based on empirical data showing that central midfield players have higher scan frequencies compared to other playing positions [24], presumably because they are (more often than players in other positions) literally surrounded by multiple sources of information (the ball, teammates, opponents, etc.), which makes constant scanning activity essential for performance. As we aimed to study an elite sample, an additional inclusion criterion was that the player had to have played in the starting 11 of their respective team for more than one game. The players had, at the time of the data gathering, started between five and 71 matches ($M$ = 38.25, $SD$ = 26.09). One additional player, who was also part of the data collection, was excluded from the analysis based on this criterion to ensure that all players were in fact elite, consistent with previous scanning studies [24, 25]. Data from those five players' eye tracking records was also used in another study, which exclusively focused on the fixations of the players [8] compared to the current study, which exclusively looks at the scanning of the players. Written informed consent was obtained by all participants prior to data collection in accordance with the General Data Protection Regulation and the Declaration of Helsinki. The study was approved by the Norwegian Centre for Research Data (NSD), reference number 52593, prior to data collection.

### Procedure

Both clubs were contacted via e-mail and telephone, and subsequent meetings took place between the clubs and the first and fourth authors. The dates for two separate data collections were agreed to by the coaching staff and the first author. Prior to the data collection, two pilot tests on elite youth players were conducted. These studies revealed the importance of attaching the eye-tracking battery in a secure and stable way and maintaining similar lighting conditions during the calibration and throughout the data sampling.

Data was collected during two 11 vs. 11 matches played with standard association football rules. One match was an internal training match within the squad, while the other was a friendly match against a local third division team. Data was collected during the competitive season of the two teams. At both matches, prior to the warm-up, the participants were each equipped with an eye-tracking device to allow them to familiarize themselves with the equipment and to ensure that a stable calibration was possible. This process lasted approximately three minutes for each participant. In total, two of the players were recorded for 20 minutes each, and two players were recorded for 10 minutes each. The difference in duration was due to (a) the match duration and (b) the duration of the fitting process. As this study does not analyze individual differences in any way, we decided to include all recorded data irrespective of duration.

### Equipment

The eye-tracking device used to register gaze behavior when performing scanning was the Tobii Pro Glasses 2 (Tobii Technology AB, Sweden). The Tobii Pro Glasses 2 is a mobile binocular eye tracker operating at 50 Hz with four built-in infrared sensors catching the movements of each eye. It also contains a high-definition camera (1920 × 1080 px, 25 fps) with a minimum of 82˚ horizontal and 52˚ vertical detection, which films the visual scenery of the user. The glasses operate with a visual span of over 160˚ horizontally and 70˚ vertically according to the Tobii documentation [26]. The visual behavior was registered and stored by the Tobii Pro Glasses Controller version 1.73.8622 on a 32 GB memory card. The memory card was localized in a recording unit strapped onto each player's shorts or back, allowing him to move freely (see Fig 1).

We also used a Panasonic AG-UX90 4K camcorder to film the match from a platform situated on the sideline approximately 5 m above the ground near the midfield line. Data from the camcorder was used to measure distances between players and the ball during scans when the ball would not be visible in the eye-tracking video. This ensured that the context could be accurately measured.

### Variables

Based on Gibson's conceptualization of exploration [13] and Jordet's [14] operational definition of an exploratory search, we defined visual exploratory scanning (scanning) as an active head and eye movement away from the ball that temporarily causes the ball to fall outside of the participant's visual field (eye-tracking camera). The player presumably performs this motion with the intention of looking for information from teammates, opponents, the referee, or space that is relevant to the development of play (see Fig 2).

Only scans that were performed during open play were analyzed, with the exception of scans that were initiated within the two seconds leading up to a set-piece being taken, as this was viewed as an important time for information gathering. Additionally, in accordance with previous studies [24, 27], scans were only measured when the participants were not in possession of the ball. All scans detected from the four players were used in the analysis, totaling 869 scans (Player 1 = 381, Player 2 = 208, Player 3 = 177, Player 4 = 103). The data collection focused on two main properties of scanning as dependent variables: scanning duration and scanning information.

**Dependent variables.** *Scanning duration* was defined as the duration of scans in centiseconds (cs), as measured by Tobii Pro Lab (centiseconds are used as the time measurement scale throughout this paper as it provided us with the most accurate description of the results). Scanning duration was measured from the first video frame in which the ball was not visible

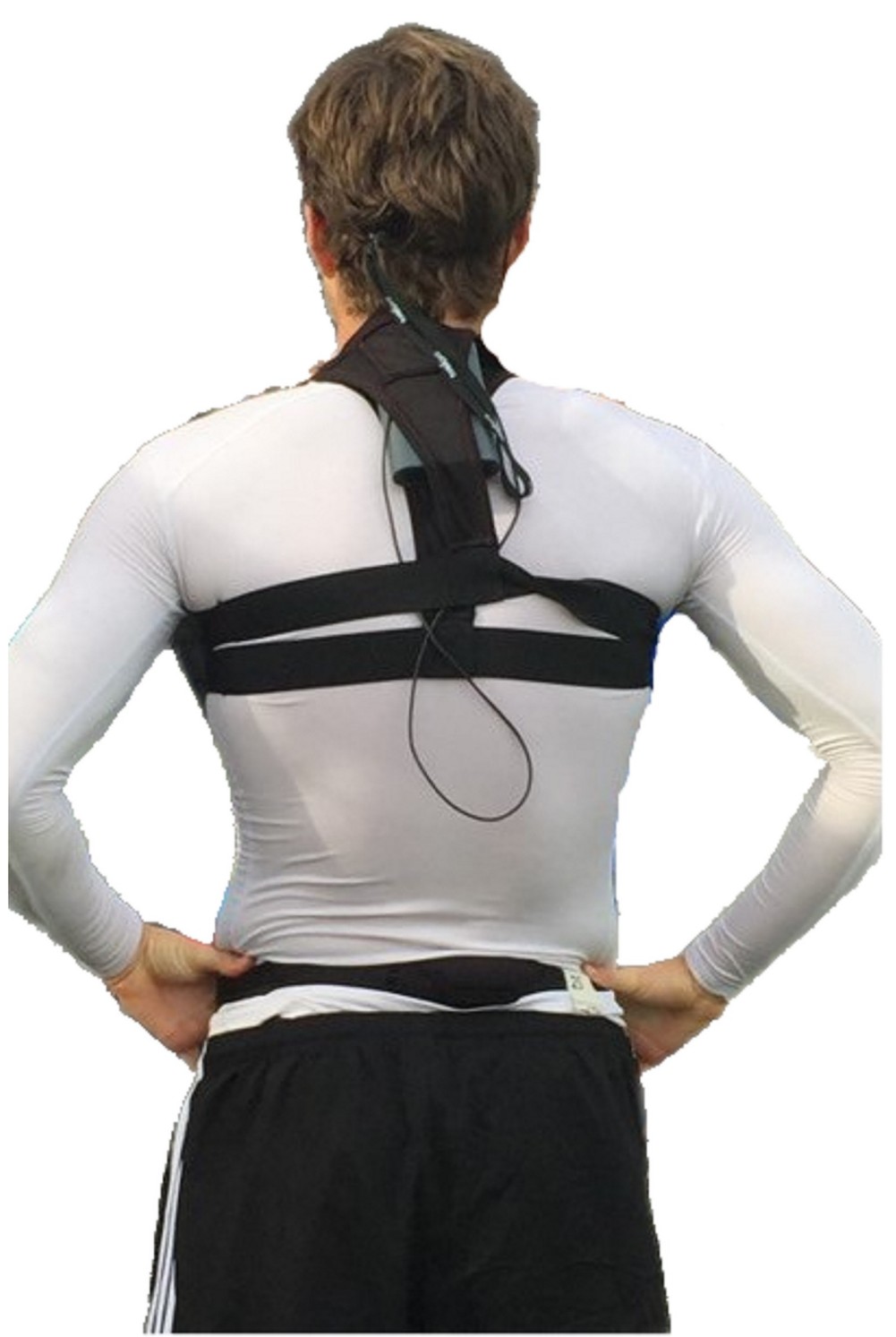

**Fig 1. The Tobii Pro Glasses 2 recording unit attached on the upper back of one of the participants.** Printed with permission.

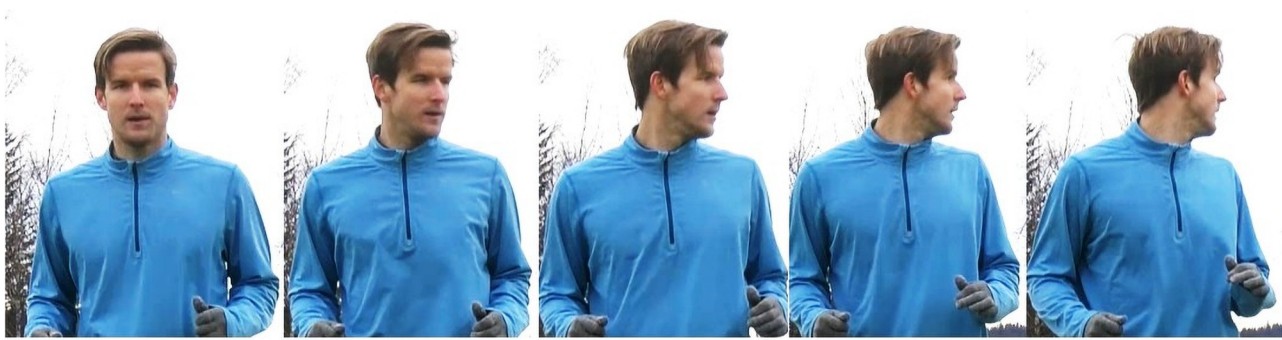

**Fig 2. Illustration of a visual exploratory scan directed from the ball's position (far left) towards information to the left of the player (far right).**

inside the eye-tracking video to the first video frame in which the ball once again became visible. This operationalization was constructed to ensure maximum objectivity when measuring the start and end of a scan. The limitations of this operationalization were (1) micro scans in which the ball does not leave the video frame (these were excluded from the analysis) and (2) most scans start a few unequal numbers of centiseconds before our measurement starts.

*Scanning information* was the collective term for the number of players (i.e., teammates and opponents, respectively) visible during the scans (both foveally and in the scene camera). Scanning information was measured in three different ways. First, the number of players inside the entire video frame during the movement phases of the scan, which was defined as the number of teammates and opponents found inside the eye-tracking video frame during the two movement phases (away from and towards the ball), was determined. This excluded the number of teammates and opponents in the video frame at the stop point of the scan. This exclusion was made in order to not retain any overlapping data points between the number of players found in the entire video frame in the different moments of the scan (movement phases and stop point). Second, the number of players inside the entire video frame during the stop point of the scan was also measured. This was defined as the number of teammates and opponents found inside the video frame at the moment in which the player had the last stop point of the scan before moving his head and eyes back towards the ball. Third, the number of players found inside the foveal circle, measured at 100% in Tobii Pro Lab, during the stop point of the scan, was also measured. The stop point video frame was the last video frame before the direction of the scan was reversed.

**Independent variables.** With regard to independent variables, we measured those that provided additional context to the scanning duration and scanning information at the exact moment of the *initiation* of the scans. Four independent variables were used to provide further context for scanning duration: control or pass, air or pitch, ball action, and the presence of fixations.

The following operationalizations were made with reference to other players, as we did not measure scanning when the participants (the players equipped with eye-trackers) had possession of the ball. *Control or pass* refers to whether the scan was initiated when a player had control of the ball (either by touching it or between touches) or when the ball was on its path from one player to another. Control was defined as having the ball close to the player's body after the initial receiving touch. *Air or pitch* refers to whether the scan was initiated when the ball was on the pitch (i.e., field) or up in the air. *Ball action* refers to the action that was undertaken with the ball at the exact moment the scan was initiated. This was divided into five categories: (a) receiving/dribbling touch, (b) during pass (the path of the pass), (c) out of play, (d) control, no touch (a player had possession of the ball, but it was between touches), and (e) moment of

pass (touch). Lastly, to measure whether players foveally fixated on an object and/or space during their scanning, we measured the *presence of fixations* using the Tobii Pro Lab fixation filter set at a 120 ms threshold [28]. This threshold is similar to other gaze behavior studies in football conducted in laboratory settings [5, 29, 30], and it is in line with the 100–200 ms thresholds that are most frequently used in gaze behavior studies [31]. However, we argue that these threshold guidelines, originated from controlled laboratory settings, may not be able to accurately capture the shorter fixations that more likely occur in unrestricted field studies such as football match play. Thus, we included a S1 Data in which the fixation detection threshold was set at 60 ms to make our data available for comparison for future analysis once a lower fixation detection threshold has been considered in the scientific community.

Additionally, two independent variables were analyzed in order to provide a scanning context in both scanning duration and scanning information: playing phase and player-to-ball-distance. The *playing phase* was split into attack and defense. Attack was operationally defined as the period when the investigated player's team had control of the ball; it ended when they lost possession to the other team, the ball went out of play, or a free kick was awarded [8]. Defense was operationally defined as the period when the investigated player's team did not have control of the ball; it ended when the opposition team lost possession to the investigated player's team, the ball went out of play, or a free kick was awarded [8]. We operationalized that a team had control of the ball when a player made two or more touches or was able to make a controlled pass or shot using his first touch. If neither team had control of the ball at the initiation of the scan, it was categorized as "other." *Player-to-ball distance* was defined as the number of meters between the analyzed player and the ball when a scan was initiated. This variable was subsequently divided into two groups: near (0–24 meters) and far (25–47 meters), based on similar previously used distinctions [5, 8].

## Data analyses

The data analysis was conducted using Tobii Pro Lab (version 1.70.8207) and a split-screen synchronization of the video from the eye tracker and video from the camcorder, which was produced using the Sony Vegas Pro 13 program, and analyzed using the program Assimilate Scratch Play (version 9.2). Each scan was analyzed frame by frame in 50 frames per second (2 cs frame interval); there were a total of 869 scans. As the HD video camera attached to the Tobii Pro Glasses 2 eye-tracker only filmed at 25 fps (4 cs frame interval), synchronizing the video with our overview video (50 fps) made it possible to register scans at a 2 cs interval. However, this resulted in a higher number of scans being registered to end during odd frame numbers because every other frame would be blurry. The analysts were instructed to be certain that the ball had re-entered the video frame before registering the end frame of a scan.

In order to assess the reliability of the data, both an intra-observer and an inter-observer test were conducted on 10% of the complete dataset. The intra-test was conducted by the second author six weeks after the initial data analysis. The inter-test was conducted by a Union of European Football Associations (UEFA) B licensed coach with a bachelor's degree in sports science, who went through an intensive one-day training period to familiarize himself with the equipment and the variables. Cohen's kappa intra-observer strength of agreement [32] was perfect for the playing phase ($k = 1$), almost perfect for control or pass ($k = .98$), scanning initiation ($k = .96$), air or pitch ($k = .92$), and fixations ($k = .94$). Similarly, the Cohen's kappa inter-observer agreement was perfect for the playing phase ($k = 1$), almost perfect for control or pass ($k = .94$), scanning initiation ($k = .96$), air or pitch ($k = .80$), and fixations ($k = .87$).

Additionally, the intraclass correlation coefficient (ICC) was applied to measure the agreement of the scale variables [32]. The intra-observer test showed very strong agreement for

player-to-ball distance (ICC = .99), teammates and opponents in the video frame during the movement phases of the scans (ICC = .97), teammates and opponents in the video frame during the stop point of the scans (ICC = .99), and teammates and opponents in the foveal circle during the stop point of the scans (ICC = .96). Similarly, the inter-observer test showed very strong agreement for player-to-ball distance (ICC = .99), teammates and opponents in the video frame during the movement phases of the scans (ICC = .96), and teammates and opponents in the video frame during the stop point of the scans (ICC = .99), as well as acceptable agreement for teammates and opponents in the foveal circle during the stop point of the scans (ICC = .78).

## Statistical analyses

Statistical tests were performed using SPSS 27.0 (SPSS Inc., Chicago, IL, USA). A Shapiro–Wilk test of normality showed that scanning duration significantly deviated from a normal distribution, W(869) = 0.74, $p < .01$, z (skewness) = 4.07, z (kurtosis) = 123.12. Consequently, we used non-parametric tests for all analyses in which scanning duration was used as a dependent variable. This included Mann–Whitney U tests for the analysis of the independent variables *control or pass* as well as *air or pitch* and a Kruskal–Wallis test for the analysis of the independent variable *ball action*. Additionally, Mann-Whitney U tests were used for *fixations in scanning*, *player-to-ball distance*, and *playing phase*. Regarding the three ways we used to measure scanning information (movement phases, stop point, and foveal circle stop point), ANOVAs were conducted for the number of players (teammates and opponents) with player-to-ball distance (near, far) and playing phase (attack, defense) as independent variables. Partial eta squares were calculated as effect size measures. The alpha level for all statistical tests was set *a priori* at α = .05.

## Results

### Scanning duration

The players in this study performed 869 scans with a mean duration of 39.65 cs (0.3965 seconds) (*Mdn* = 34 cs, *SD* = 28.42, Max = 328 cs, Min = 2 cs). As depicted in Fig 3, 90.3% of all scans performed ranged from 2 to 66 cs, and the most common duration was 26 cs (*n* = 95).

### Scanning duration and ball context

Of the 869 analyzed scans, 835 were performed when the ball was on its path between two players (pass) or when a player had control of the ball. Initial analyses using a Mann–Whitney U test revealed that players had longer scanning durations when the ball was on its path (pass) (*M* = 44.32 cs, *SD* = 30.62, *n* = 433) than when the ball was under control (in possession) by a player (*M* = 34.61 cs, *SD* = 24.95, *n* = 402), *U* = 67772, *z* = -5.54 $p < .001$, $\eta_p^2$ = .04. In order to analyze the duration on scans initiated during different contexts further, we divided ball action into (a) receiving or dribbling touch; (b) during pass (the path of the pass); (c) out of play; (d) control, no touch (a player had possession of the ball, but it was between touches); and (e) moment of pass (see Fig 4).

A Kruskal–Wallis test revealed significant differences between the groups, *H*(4) = 41.20, $p < .001$. Post hoc, pairwise comparisons with adjusted *p*-values showed that significantly shorter scanning durations occurred when the ball was controlled by a player (without him touching it) compared to when the ball was on its path between two players after an executed pass ($p < .001$). No significant differences were found between the other groups. However, a

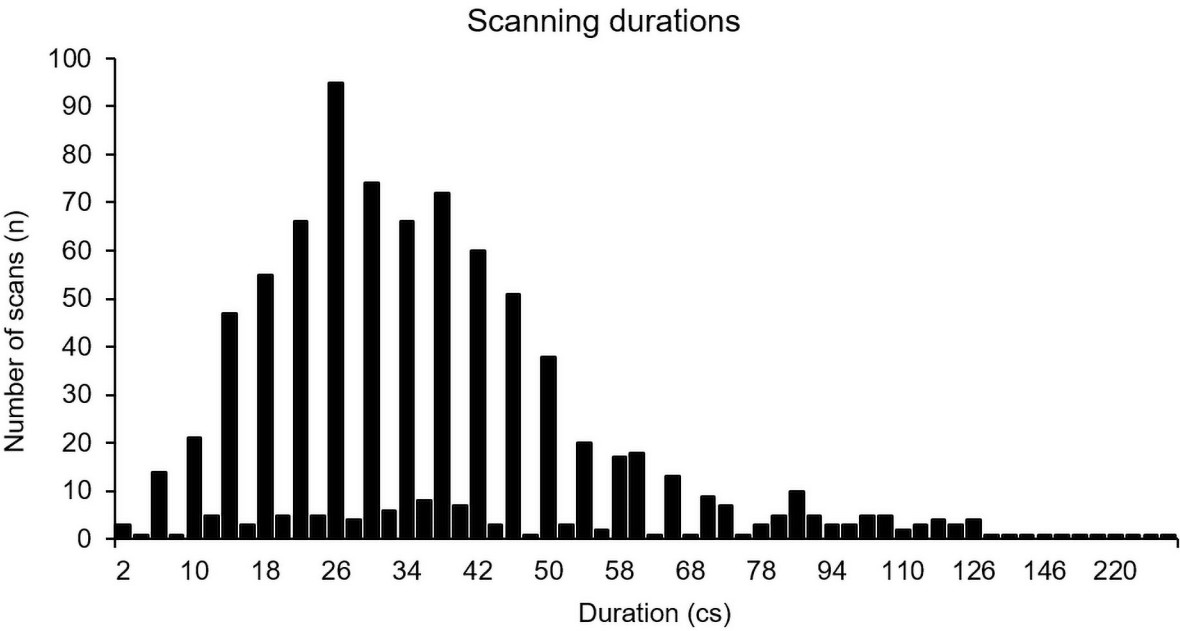

**Fig 3. Number of scans as a function of different durations.**

trend was found suggesting that longer scans occurred during a receiving or dribbling touch compared to when the players had control of the ball without touching it ($p = .062$).

Additionally, we looked at the duration of scans initiated when the ball was up in the air compared to when it was on the pitch. A Mann–Whitney U test revealed a significantly higher

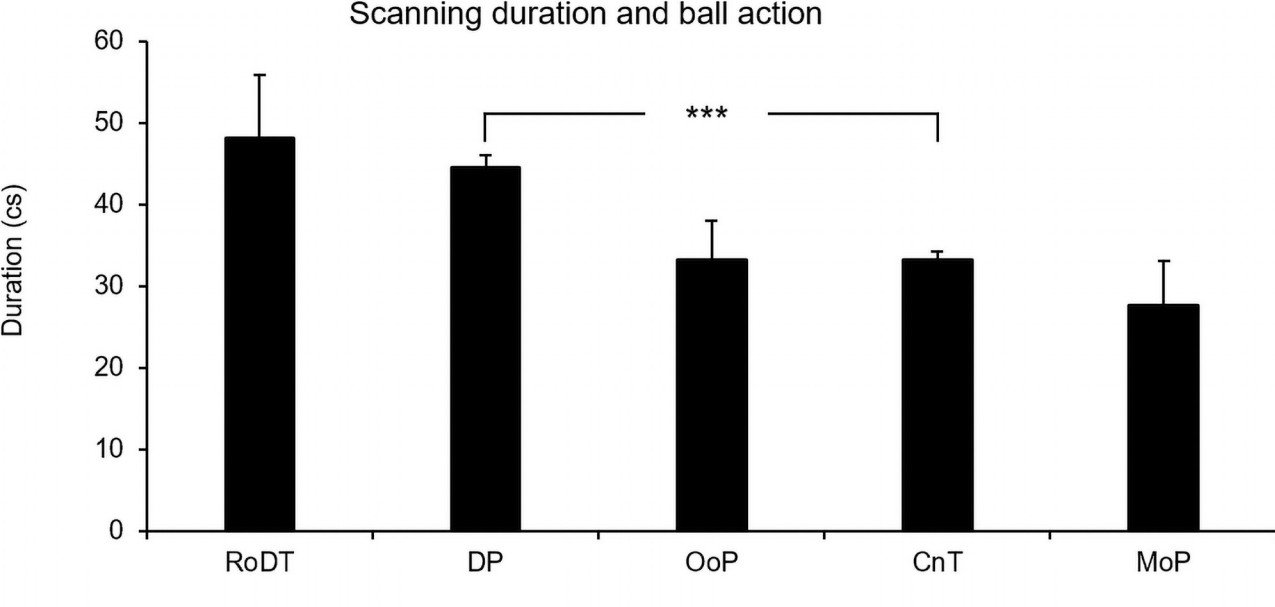

**Fig 4. Means and standard errors of scanning duration during different ball actions: Receiving or dribbling touch (RoDT); during pass (DP); out of play (OoP); control, no touch (CnT); and moment of pass (MoP).**

scanning duration when the ball was in the air ($M$ = 45.24 cs, $SD$ = 35.45, $n$ = 239) compared to when the ball was on the pitch ($M$ = 37.54 cs, $SD$ = 24.95, $n$ = 630), $U$ = 67343, $z$ = -2.41, $p$ = .016, $\eta_p^2$ = .01.

## Scanning duration and the presence of fixations

Of the 869 scans analyzed in this study, only 20 (2.3%) involved a fixation (Player 1 = 5, Player 2 = 10, Player 3 = 3, Player 4 = 2) when using a fixation detection threshold of 120 ms. Initial analyses revealed longer average durations for scans that involved fixations ($M$ = 97.10 cs, $SD$ = 57.12, $n$ = 20) compared to scans that had no fixations present ($M$ = 38.30 cs, $SD$ = 25.96, $n$ = 849). A Mann–Whitney U test showed that scans that included fixations were significantly longer than scans that did not include any fixations, $U$ = 2116, $z$ = -5.76 $p$ < .001, $\eta_p^2$ = .04.

## Scanning duration, player-to-ball distance, and playing phase

To test the relationship between scanning duration, playing phase, and player-to-ball distance, we conducted separate Mann–Whitney U tests, using scanning duration as the dependent variable. The Mann–Whitney U tests revealed that there was no difference in duration between when the scans were conducted in the near (0–24 m) ($M$ = 39.03 cs, $SD$ = 27.37, $n$ = 670,) and far (25–47 meters) conditions ($M$ = 39.39 cs, $SD$ = 25.13, $n$ = 191,), $U$ = 61648, $z$ = -.77, $p$ = .440, $\eta_p^2$ < .01. Furthermore, no difference in duration was found between defense ($M$ = 41.15 cs, $SD$ = 35.34, $n$ = 341) and attack ($M$ = 38.68 cs, $SD$ = 22.86, $n$ = 528), $U$ = 88371, $z$ = -.46, $p$ = .65, $\eta_p^2$ < .01 (see Fig 5).

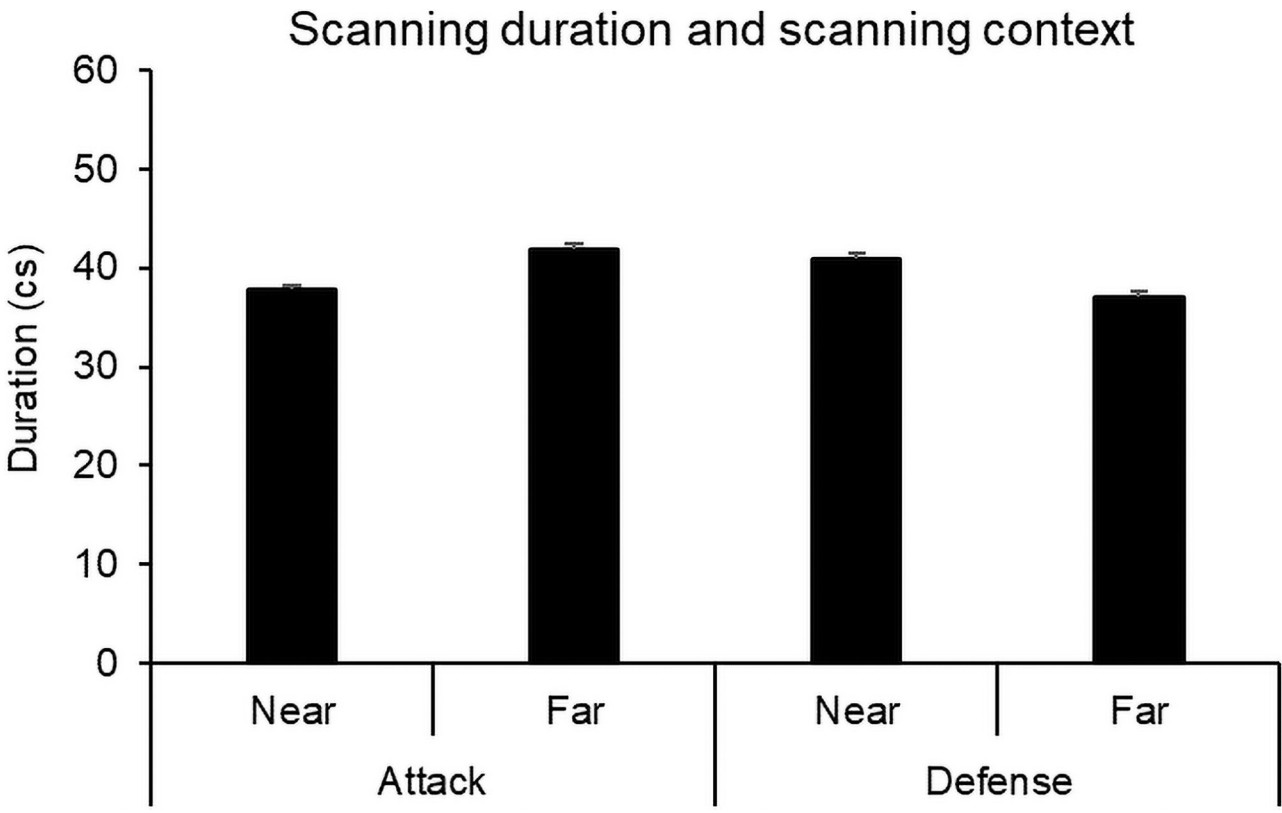

**Fig 5. Means and standard errors of scanning duration as a function of playing phase (attack, defense) and player-to-ball distance (near, far).**

## Scanning information, player-to-ball distance, and playing phase

To assess scanning information, the first set of analyses investigated the number of teammates and opponents inside the video frame during the scans. Fig 6 compares the summary statistics of teammates and opponents according to the three ways of measuring scanning information we used in this study: movement phases ($n_{scans}$ = 867), stop point ($n_{scans}$ = 867), and foveal circle stop point ($n_{scans}$ = 758). From the graph below (Fig 6), we can see that, in the movement phases of the scans, the players most often had zero teammates and opponents inside the video frame. This result should be seen in light of our operationalization of the movement phase which excluded all players that were visible inside the stop point of the scan. Furthermore, the players never had more than seven teammates in their video frame; they did have both eight and nine opponents in their video frame, although this happened infrequently. In contrast, the highest count found at the stop point of the scans was one to three players for both teammates and opponents.

Lastly, compared to the movement phases and the stop point, the foveal circle stop point of the scans showed a lower number of players (see Fig 6). For the foveal circle stop point, zero teammates and opponents were most frequently found. No more than two teammates and three opponents were inside the foveal circle during the stop point of the scans.

To assess how scanning information changes as a function of the playing phase and player-to-ball distance, a three-way ANOVA of the playing phase (2) × player-to-ball distance (2) × number of players (2), with repeated measures on the last factor, was conducted separately for the movement phases, stop point, and foveal circle stop point. For the movement phases, the analysis revealed a significant main effect for the playing phase, $F(1, 857) = 29.23$, $p < .001$, $\eta_p^2 = .03$, a significant main effect for number of players, $F(1, 857) = 28.68$, $p < .001$, $\eta_p^2 = .03$, and an interaction between the playing phase and the number of players, $F(1, 857) = 8.71$, $p = .003$, $\eta_p^2 = .01$. No other main effects or interaction effects were found.

These results show that during the movement phases of the scans, more players were found inside the video frame during attack than defense and that there were, in total, more opponents than teammates inside the video frame during the movement phases of the scans. More precisely, while in defense, no differences could be found between the amount of opponents and teammates in the video frame. In attack, there were more opponents than teammates inside the video frame during their scanning behavior (see Fig 7).

Similarly, the analysis for the stop point revealed a significant main effect for the number of players, $F(1, 857) = 50.39$, $p < .001$, $\eta_p^2 = .06$, and the interaction of the playing phase and number of players, $F(1, 857) = 31.95$, $p < .001$, $\eta_p^2 = .04$. However, there were no main effects for the playing phase, $F(1, 857) = 0.10$, $p = .747$, $\eta_p^2 < .01$, or player-to-ball distance, $F(1, 857) = 1.02$, $p = .204$, $\eta_p^2 < .01$, nor was there any other interaction. The finding that more opponents than teammates were found inside the video frame during the stop point only occurred during the attack phase (see Fig 8).

For the foveal circle stop point, a main effect for the playing phase, $F(1, 747) = 7.32$, $p = .007$, $\eta_p^2 = .01$, and a significant main effect for the number of players were found, $F(1, 747) = 4.28$, $p = .039$, $\eta_p^2 = .01$. No other main effect or interaction was found. Again, more opponents than teammates were found inside the foveal fixation circle; however, there were significantly more players found in defense compared to attack (see Fig 9). This result was the opposite of what was found in the movement phases.

## Discussion

This study aimed to explore the scanning behavior of four elite football midfield players in 11 vs. 11 match play. More specifically, we wanted to examine the duration and information of

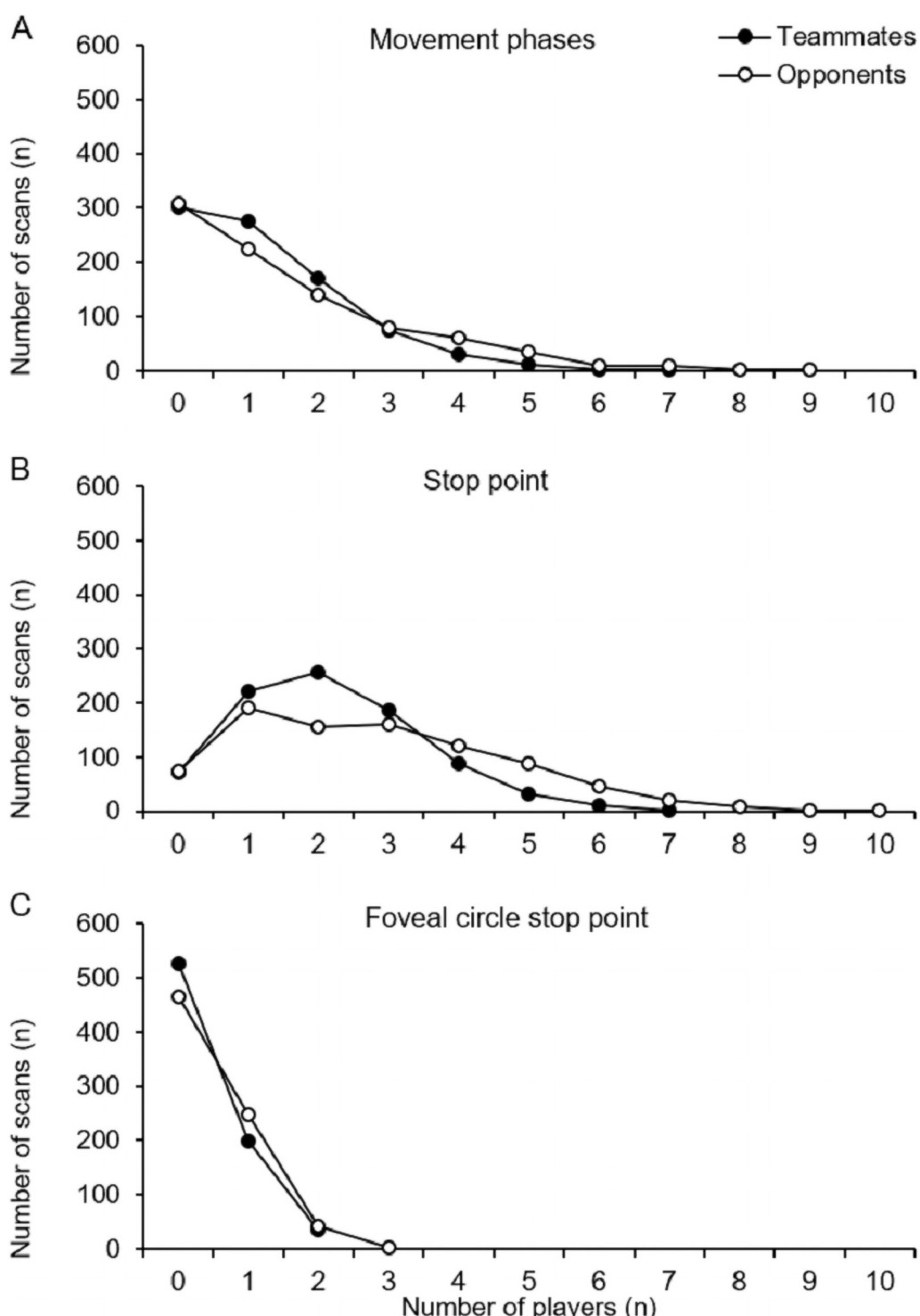

**Fig 6. Number of scans on different numbers of opponents and teammates found in the video frame during the movement phases (A), the stop point (B), and the foveal circle stop point (C).**

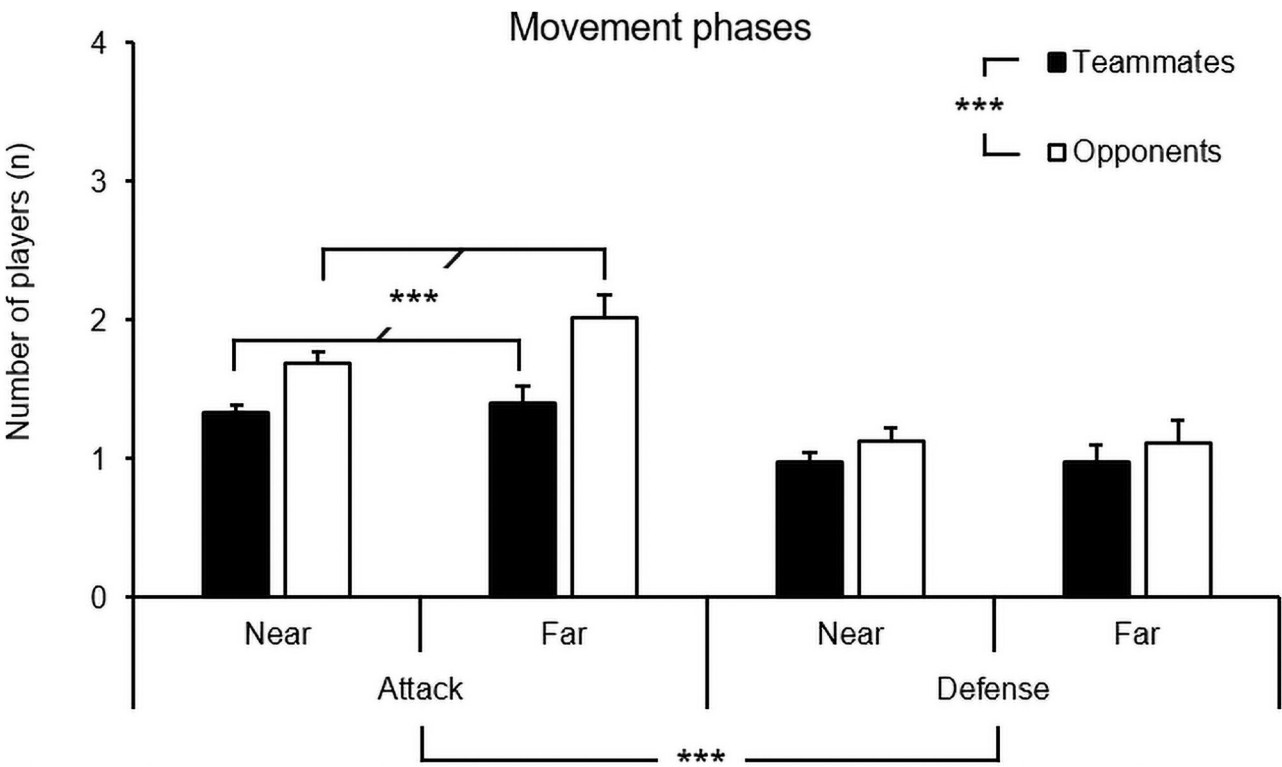

**Fig 7. Means and standard errors of the number of teammates and opponents during the movement phases of the scans as a function of playing phase (attack, defense) and distance (near, far).**

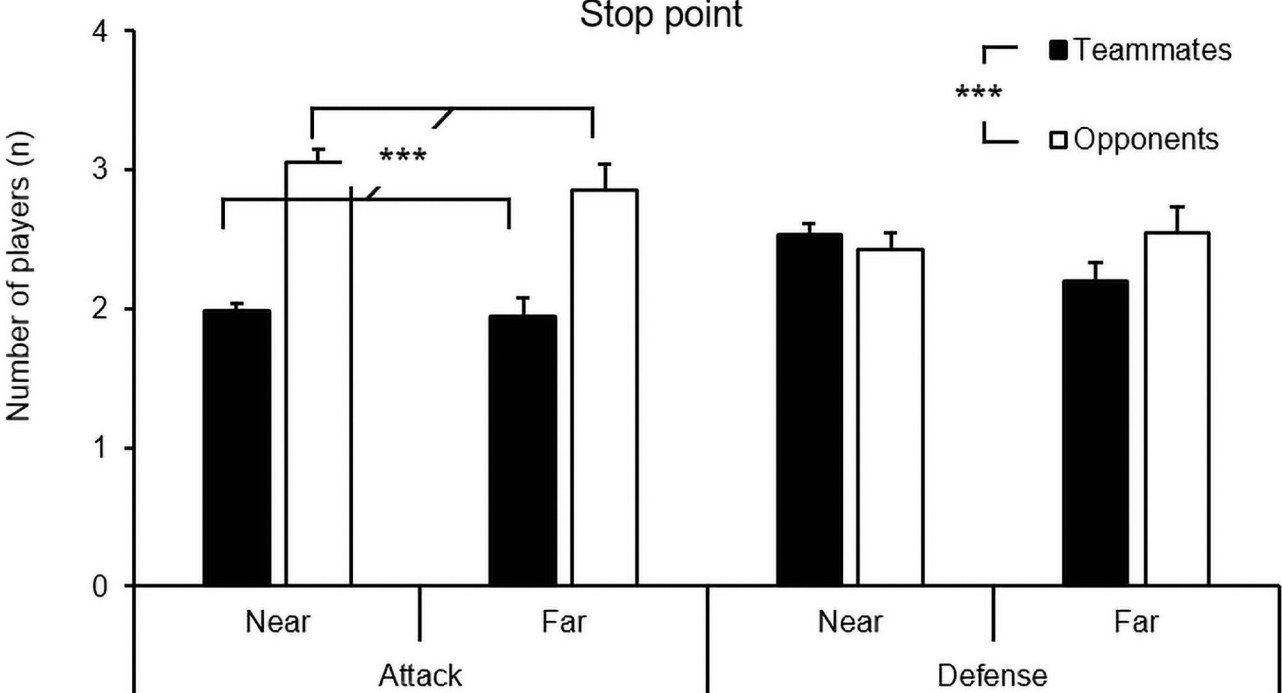

**Fig 8. Means and standard errors of the number of teammates and opponents within the entire video frame at the stop point of the scans as a function of the playing phase (attack, defense) and distance (near, far).**

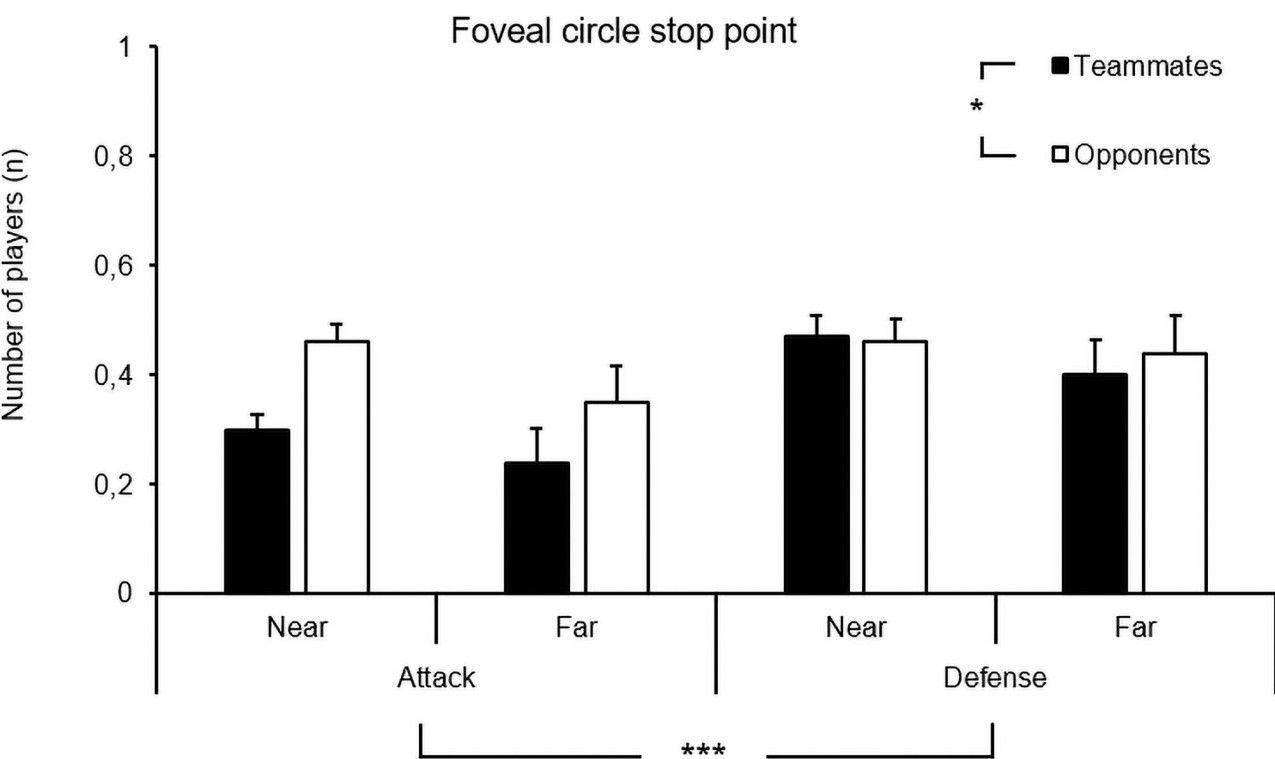

**Fig 9. Means and standard errors of the number of teammates and opponents in the foveal circle of the stop point of the scans as a function of the playing phase (i.e., attack, defense) and distance (i.e., near, far).**

scanning in different contexts using modern mobile eye-tracking equipment. The results of this study indicate that (a) the action undertaken with the ball at the moment of scanning initiation influenced scanning duration; (b) playing phase and player-to-ball distance influenced the number of teammates and opponents inside the video frame during scanning; (c) very few scans involved fixations; (d) based on our operationalizations, the different parts of a scan revealed different detectable visual information; and (e) scanning duration is not influenced by player-to-ball distance and playing phase. Given that our findings are based on a limited number of participants (four) from a homogenous population (elite midfielders), the results from our analysis, which will now be discussed, should be treated with considerable caution.

The first main and novel finding of this study is that scanning duration is influenced by the action undertaken with the ball at the moment a scan is initiated. In particular, the results showed that the players performed significantly longer scans when (a) the ball was in the air rather than on the pitch and (b) when the ball was passed between two players rather than when players had control of the ball but did not touch it (between touches). Both results suggest that the players in this study were more inclined to scan for longer durations, thus allowing them to gather more information when the future position and direction of travel of the ball could be more precisely anticipated. This main result also substantiates the notion from ecological psychology that perception and action are closely coupled (e.g., [13]), by showing how scanning behavior changes based on different action requirements and game situations.

In football, when the ball is being passed along the ground, the path and, consequently, the probable destination of the ball can be anticipated by skilled players (even more so when the pass is made through the air, where there is no one to intercept the ball). Performing scans with longer durations is logically similar to performing scans with bigger head

excursions. Larger head excursions have been found to be indicative of better subsequent performance with the ball, such as faster passing responses [22] and the ability to switch play and turn with the ball [20]. It is therefore plausible that the players in this study were able to detect, based on the action on the ball, when it is possible to look away from the ball for longer durations (e.g., when the ball is in the air or the ball is on its path from one player to another) and when situations were more uncertain, requiring players to return their attention swiftly to the ball to achieve situational control (e.g., when the player has control of the ball but is not touching it). However, no previous studies have examined the durations of scans or head turns in relation to the action undertaken with the ball. Hence, these assumptions should be cautiously interpreted.

Second, the results showed no significant difference in scanning duration between the near (0–24 m) and far (25–47 m) distance conditions, suggesting that scanning behavior is not impacted by the player-to-ball distance. This finding was somewhat unexpected because previous studies on football with matching distance classifications have found that players' fixation durations are highly influenced by player-to-ball distance [5, 8, 30]. Although those studies did not measure scanning, they did look at sources of information for football players, making their results somewhat comparable to ours.

Similarly, our findings revealed no statistical difference between scanning duration in attack and defense. This finding is in agreement with McGuckian et al.'s [22] findings, which showed that there was no difference in head turn excursion between players in defense and attack (except by the player in possession) in both the vertical and horizontal pitch dimension analysis. This finding, while preliminary, suggests that the playing phase does not influence scanning duration, meaning that players perform scanning in a similar way in both attack and defense.

Our third main finding was that fixations were almost non-existent during the players' visual exploration, occurring in only 2.3% of the scans and only in scans with long durations. This result is highly interesting, as fixation properties are the most investigated aspect of gaze behavior in sports research to date [3]. The absence of fixations implies that players, when scanning, do not need to foveally fixate on the surrounding objects and spaces in order to acquire sufficient information for guiding their next action. This result may partly be explained by the fact that the adopted 120 ms threshold for fixation detection used in this study is not well applicable for real-world research, in which unstable and rapid movements occur all the time [8]. Thus, these data need to be interpreted with caution and show the need for investigating fixations during scanning in football further, preferably with a lower fixation detection threshold. A suggested threshold of 70–80 ms would probably include more fixations for the analyses whilst maintaining a sufficient duration to account for the uncertainty of saccadic suppression (where information cannot be processed) [33] (see S1 File for the analyses using a lower fixation detection threshold of 60 ms). Nevertheless, this key finding supports Gibson's assumption that human perception should not be equated or compared to pictorial perception [13], during scanning.

Our fourth main finding was that, somewhat surprisingly, scanning durations were lower than expected. This result could be partly explained by our operationalization of a scan in which we started measuring the duration at the moment the ball left the video frame. Of all the 869 scans, 90.3% lasted for 66 cs or less. This result, coupled with the result showing that very few scans involve fixations, support the notion that visual perception during scanning occur between the individual and the surrounding light and not the retinal image. This result is in agreement with Jordet's [14] study of three elite midfielders, which found that so few scans lasted in excess of one second, referred to as long searches, that the results became inconsequential. In comparison, a recent study on scanning behavior in football using VR simulations

focused on scans that lasted longer than one second, which the authors referred to as long exploratory activity [34]. While VR has the potential to create realistic simulations, our results show that scanning usually lasts much less than one second. Thus, once more, these results show that researching a real-world phenomenon, such as scanning, is problematic once we move outside of the actual performance context [35].

So far, this discussion has focused on scanning duration. The following section discusses the information that the players had inside their video frame when scanning. More specifically, the number of teammates and opponents visible during the scans in both the foveal vision and the scene camera. In the current study, the results from the foveal circle in the stop point of the scans showed that players had significantly more players inside their video frame in their foveal vision in defense than in attack. This result was not found in the movement phases or the entire stop point. Although eye-tracking devices cannot reveal where the user's attention is at a certain point in time [36], the foveal eye position is often similar to or the equivalent of focus of attention [37]. Whether this attentional process relates to the conscious or self-organizing tendencies of movement control remains unclear and may have important implications for practice [38]. Furthermore, this finding suggests that the players in this study were more concerned with looking for the positioning of teammates and opponents in defense. In attack, they focused more on the open spaces that they could either exploit themselves, or use to play a pass into, if they received the ball. Whether this is related to strategy, shared intentionality, or the more generic properties of their skilled behavior remains unclear. One possible explanation is that the affordances (opportunities for action) available for the players change as a function of the playing phase because of the more dynamic structure of the attack compared to defense [39]. In attack, players might be looking for the spaces and gaps that are always opening and closing [39], whereas in defense, the play might be more structured, allowing the players to focus more closely on the player in possession.

Another important finding was that there were more visible players (both teammates and opponents) in the stop point of the scans than in the movement phases of the scans (away from the ball and towards the ball). It is, therefore, probable that the football players moved their heads and eyes until they arrived at a specific point where they wished to gather perceptual information before returning their attention to the ball. However, these differences can be explained in part by the fact that many scans were done with small head excursions; thus, the area in which the head was traveling in the movement phases was smaller than the area of the video frame at the stop point for these specific scans. Hence, these data must be interpreted with caution as they are a product of the operationalizations that was used in the current study.

In our study, we found that the players detected more opponents than teammates during their scans. This finding appears to be well substantiated: we found the same result in all phases of the scans. Thus, it is possible that the players in this study were more concerned with the movement and positioning of opponents than with their teammates when it came to gathering surrounding information. However, there are also two likely natural causes for these differences: (a) in football, it is possible to detect 11 opponents but only 10 teammates, and (b) the midfield players in this study, based on their pitch position, would often scan in the attacking direction where the opponent most often has numerical superiorities.

## Limitations

Several limitations of this study must be acknowledged. First, although designs with few participants have been found to have high power and yield robust results [40], the study's limited sample size, using exclusively midfielders, does not allow us to draw inferences regarding

statistical generalizability. Second, the study design did not measure how scanning influenced players' decision-making and performance, limiting the results to descriptive accounts. Third, our operationalization of a scan meant that small head movements when the ball was still visible inside the video scene camera (e.g., on the edge of the screen) were not included as a scan. In this way, we ensured that all scans were, in fact, scans. However, this also meant that some small excursion scans might have been excluded from the analysis. Fourth, the fixation detection threshold of at least 120 ms adopted in the current study has been the standard for gaze behavior research in sports conducted in both laboratory (e.g., [28]) and field-based studies (e.g., [36]) for decades. However, this threshold originated from laboratory studies in controlled settings with little to no movement [41]. Hence, adopting a lower threshold for fixation detection will include more fixations in the analysis and, thus, could be a better approach to combine measures of scanning and fixations in future real-world sports studies.

## Practical applications

We believe that our findings, although exploratory and limited, may be useful to coaches who wish to improve their players' scanning behavior. Research has shown that although highly qualified coaches believe that scanning is vital for football performance, they find it difficult to deliver training on scanning [42]. Most of the scans in this study were shorter than 0.5 seconds and did not involve fixations. This suggests that coaches, in line with our results, could consider creating exercises in which scans need to be performed quickly, in a dynamic affordance-rich environment. For the same reason, coaches should also limit their use of non-contextual information that players need to fixate on during their scans in order to perform a task, such as counting the number of fingers the coach is presenting or reading a number on a sign. Coaches should likely instead strive to include a more representative perception–action link when training scanning skills in players [43], which entails including information that players typically need to act upon during match play. Furthermore, inspired by our results, the scans conducted should probably be linked to a subsequent decision and action response, such as turning, passing, or directed dribbling.

Our combined findings that players had more opponents than teammates inside the video frame and that these numbers changed according to playing phase, during their scanning behavior, may imply that coaches should create practices that involve the detection of that particular type of information in order to be representative. Therefore, coaches should limit the delivery of unopposed exercises where football actions are made in the absence of opponents and/or playing phases, or at the very least coaches could create exercises where players need to perceive information coming off opponents to solve the exercise efficiently.

However, although the findings represent real world elite scanning behavior, it should not necessarily be confused with optimal behavior across different age groups and skill levels. Hence, with a small number of participants analyzed in a relatively small time period, caution must be applied to both the results and the practical applications, as the findings might not be representative of other populations in different football contexts.

## Future research

The exploratory nature of this study highlights important areas for future research. Overall, future research should attempt to answer questions that originated from this study and build on the research method used to provide more knowledge of this under-investigated research area. First, studies should investigate differences in scanning duration and information between playing positions. We hypothesize that both the duration and information of scanning will be different across playing positions based on the different contextual limitations and

performance tasks of these players. Second, studies should investigate the same differences in different age groups, genders, and skill levels. Third, studies should explore how different types of scans, such as scanning for orientation and scanning for action specification [44], influence behavior and performance because this might bring forward important practical implications. Finally, and most importantly, future research should aim to uncover *why* football players scan the way they do. Theory-driven research and mixed-method design that combines the eye tracking of players in 11 vs. 11 match play and subsequent game analysis interviews with the players may provide unique insights into whether scanning can be attributed to conscious or unconscious behavior.

## Conclusion

The present study was designed to explore the duration and information of scanning in actual football match play at the elite level. The study findings suggest that the duration of scanning is influenced by the context of the ball as well as the action undertaken on the ball at the moment the player decides to scan. Furthermore, scanning duration does not seem to be influenced by playing phase or player-to-ball distance. The most surprising finding to emerge from this study is that only 2.3% of scans included fixations. This result can be partly explained by the adopted laboratory fixation detection threshold of 120 ms, which might be too high for unstable real-world research. However, it also implies that players, when performing scans during match-play, do not need to foveally fixate on surrounding information in order to obtain sufficient information for performing their football actions. Furthermore, this study has shown that different parts of the scans show different types of information and that, in general, the players had more opponents than teammates inside their video frame during their scanning behavior. Hence, the scanning analysis in this study has extended our knowledge of how elite players explore their surroundings to gather information that is essential to their performance.

## Supporting information

**S1 Data.**
(SAV)

**S1 File.**
(SAV)

## Acknowledgments

The authors wish to extend a massive thanks to the clubs and players for their participation in this study. They also wish to thank Jørgen Bjørn, who conducted the inter-reliability analysis.

## Author Contributions

**Conceptualization:** Karl Marius Aksum, Geir Jordet.

**Data curation:** Karl Marius Aksum, Lars Brotangen, Lukas Magnaguagno, Geir Jordet.

**Formal analysis:** Karl Marius Aksum, Lars Brotangen, Lukas Magnaguagno.

**Investigation:** Karl Marius Aksum, Lars Brotangen.

**Methodology:** Karl Marius Aksum, Lars Brotangen, Christian Thue Bjørndal.

**Project administration:** Karl Marius Aksum, Geir Jordet.

**Software:** Karl Marius Aksum, Lars Brotangen, Lukas Magnaguagno.

**Supervision:** Karl Marius Aksum, Geir Jordet.

**Validation:** Karl Marius Aksum, Lars Brotangen, Christian Thue Bjørndal.

**Writing – original draft:** Karl Marius Aksum, Lars Brotangen, Christian Thue Bjørndal, Lukas Magnaguagno, Geir Jordet.

**Writing – review & editing:** Karl Marius Aksum, Christian Thue Bjørndal, Lukas Magnaguagno, Geir Jordet.

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
