## [Decision Letter · Decision Letter 0]

22 Jan 2021

PONE-D-20-37282

Scanning activity of elite football players in 11 v 11 match play: An eye-tracking analysis on the duration and visual information of scanning

PLOS ONE

Dear Dr. Aksum,

Thank you for submitting your manuscript to PLOS ONE. After careful consideration, we feel that it has merit but does not fully meet PLOS ONE’s publication criteria as it currently stands. Therefore, we invite you to submit a revised version of the manuscript that addresses the points raised during the review process.

We look forward to receiving your revised manuscript.

Kind regards,

Greg Wood, PhD

Academic Editor

PLOS ONE

Journal Requirements:

2. We noted in your submission details that a portion of your manuscript may have been presented or published elsewhere.

"Figure 1 has been published recently in the article "What Do Football Players Look at? An Eye-Tracking Analysis of the

Visual Fixations of Players in 11 v 11

Elite Football Match Play" in Frontiers in Psychology. This is a publicly available figure that Tobii uses in ther manual."

Please clarify whether this publication was peer-reviewed and formally published. If this work was previously peer-reviewed and published, in the cover letter please provide the reason that this work does not constitute dual publication and should be included in the current manuscript.

4. We note that Figure 2 includes an image of a participant in the study. 

Reviewers' comments:

Reviewer's Responses to Questions

**Comments to the Author**

1. Is the manuscript technically sound, and do the data support the conclusions?

Reviewer #1: Partly

Reviewer #2: Yes

2. Has the statistical analysis been performed appropriately and rigorously? 

Reviewer #1: Yes

Reviewer #2: No

3. Have the authors made all data underlying the findings in their manuscript fully available?

Reviewer #1: No

Reviewer #2: Yes

4. Is the manuscript presented in an intelligible fashion and written in standard English?

Reviewer #1: Yes

Reviewer #2: Yes

5. Review Comments to the Author

Reviewer #1: Congratulations to the authors for completing this research. There has been a real need to combine eye tracking and scanning research, and the approach the authors have taken is an excellent step in this endeavour. Overall, I appreciate the amount of work that the authors have put into this research. I believe the research provides some valuable insight into scanning, but still requires major work in ensuring the value of the research is clearly communicated. I encourage the authors to continue to develop this manuscript, and I have some comments below that may help with this. Again, I would like to congratulate the authors on this work so far. I look forward to reading a revised version.

1. The study presents the results of primary scientific research.

- Yes

2. Results reported have not been published elsewhere.

- As far as I am aware, the results have not been reported elsewhere.

3. Experiments, statistics, and other analyses are performed to a high technical standard and are described in sufficient detail.

- Overall, the design is clearly described and appears to have been performed to a high standard. I have a couple of points that I am interested to hear a rationale or further explanation from the authors on, outlined below.

- On line 66, the authors report that Aksum et al. found fixation durations to be shorter in the field than in the laboratory. Given this, I am interested to know why the authors chose a 120ms duration (L240) to define a fixation in the current study, and if this may have contributed to the low number of fixations detected during scans.

- L127-130, while I understand this rationale, it suggests that scanning is not essential for performance for other playing positions, which seems like a difficult position to justify. Would it be appropriate to also say that central midfield players were selected as it is more often relevant for them to scan the 360-degree environment, therefore increasing the available dataset?

- L179-241, for clarity, I would encourage the authors to define the variables of interest in a table or to demarcate the variables with subheadings. Further, please clearly define all variables (e.g. L239, “control, no touch”, is not clear what this could mean). It would also be useful to clearly outline which are independent and dependent variables to assist with comprehending the statistical analyses and results.

- Figure 3, are the authors able to provide some insight into why some durations (e.g. 26, 30, 34, etc.) occur much more often than other durations (e.g. 28, 32, 36, etc.)? It could be useful to provide this explanation within the text of the article.

- Figure 4 caption, please indicate what the error bars are a measure of (SD, SE, etc.?). Please also do this for other figures where relevant.

- L310-313, the authors here indicate a significant difference, but earlier state that it was not found to be relevant in providing context (L229-231). Can the authors please clarify this?

- L327-331, for clarity, please do not indicate that there was a difference between defence and attack. Instead, the phrasing used in lines 324-326 would more clearly outline the non-significant result.

- Figure 6, at first it seemed odd to me that the movement phase would have a higher number of scans with zero or one teammate/opponent than at the stop point. My reasoning was that the movement phase must include at least the number of players as the stop point, because the movement phase includes either side of the stop point. I then re-read the definition of the movement phase and noticed that the movement phase excludes players that are counted in the stop point variable (L206-207). Can the authors please provide a rationale for this? Operationally, this does not make sense to me. Teammates/opponents that are visible during the movement phase do not become invisible because the player stops moving their head, so it seems logical to include these players in both the movement phase and stop point phase variable. This would mean that for each scan, the movement phase should have at least the number of teammates/opponents as the stop point, but I do not see this as an issue. Following the definition of the movement variable (i.e. stop point players excluded), I would think that the players that are included in the foveal circle variable should be excluded from the stop point variable (but to be clear, I don’t think this would make sense either). I apologise if I am missing something clearly obvious with this comment, I will be happy to hear a good reason for this decision by the authors.

- L357-360, I feel this extra information about the analysis would be better suited in the statistical analysis section. In doing so, please also include the level names in brackets – e.g. playing phase (2 levels: attack, defence).

- L387-389, this text is repeating what was stated earlier (L357-360), so can safely be deleted. Otherwise, it suggests that a separate analysis was being conducted, which I don’t believe is that case.

4. Conclusions are presented in an appropriate fashion and are supported by the data.

- L407-408, d) maybe, but this is dependent on how these different parts are defined in the study (as above). For the player, all the visual information in the foveal circle are detectable in the stop point, and all the visual information in the stop point are detectable in the movement phase.

- L421-431, I’m not sure I follow the logic or relationship in this argument. Can the authors more strongly put forward their point in this paragraph? Yes the ball path is more predictable when it is being passed between players, but there was a difference between passes on the ground and the air, why could this be? Yes, longer scans are operationally similar to larger excursion scans, but how does the referenced literature relate to the statements from L426 onwards?

- L468-478, this is a fair point and I agree with the conclusion, however, I do also again wonder about the influence of how scans are defined and the influence this might have on the outcome. For example, a scan does not start until the ball is out of the video frame (according to the definition), however the head movement before the ball is out of the frame is still part of a scanning movement. If this were included in the definition, the duration of scans would be longer. I’m not necessarily suggesting the definition of scans needs to be changed, but perhaps this is worth a mention somewhere?

- L489-499, I really like this discussion.

- L500-510, this one I don’t agree with so much. A limitation with the definition is alluded to at the end of this paragraph. I will be interested to hear the authors thoughts on my previous comments related to this.

- L513-515, this should maybe be phrased as ‘possible’ rather than ‘likely’, particularly given that the authors provide two very valid reasons as to why more opponents would be detected, and neither of these reasons suggest that it is the players intention to try to perceive more opponents.

- L520-532, brilliant, I love this advice!

- L533-536, could this application be strengthened by including the findings relating to attack and defence? For example, players seem to detect different information during these phases, and this information is only created because there are opponents and teammates to be detected. Therefore, unopposed exercises should be limited.

- L541-549, there are potentially some other limitations that would be good to include here. I would encourage the authors to outline other limitations (e.g. the minimum duration of fixation used) in full in order to assist future investigations in overcoming any possible limitations.

- L550-560, I agree that these are good ideas. I wonder if the authors could expand a little on why these should be future investigations. Are they particularly under investigated? Will they bring about important practical applications? Will they answer questions that came up in the current study?

5. The article is presented in an intelligible fashion and is written in standard English.

- Overall, the article is well written and easy to comprehend. Complex aspects are described clearly, allowing for sufficiently clear interpretation of the article throughout. There are a couple of instances that I believe the clarity could be improved, which are outlined below for the authors to consider.

- L84-85, I feel the wording of this sentence gives the impression that affordances are only relevant for on-ball activity. This is not correct from the broader theoretical standpoint, but also in this paper which investigates both attack and defence. To clarify this sentence, I feel it would be useful to say something like “In football, affordances involving interaction with the ball rely heavily…”.

- L88-90, I feel the authors could be more direct with the explanation of the ecological approach here. Under an ecological approach, perception-action couplings are context specific (not implied), and therefore they need to be studied in the performance environment to be explained.

- L91, I’m not sure this sentence is very clear. Would it suffice to just say that “Visual scanning has been analysed…”?

- L98, should this be scanning frequently, not frequency? If the former is used, this will explain the relationship (positive effect). With the latter, it is not clear (at least, it could be more clear) if it is a high or low frequency that has the positive effect on passing performance.

- L429, ‘tits’, a small typo, I’m assuming.

6. The research meets all applicable standards for the ethics of experimentation and research integrity.

- The research appears to meet ethical standards.

7. The article adheres to appropriate reporting guidelines and community standards for data availability.

- The authors report that data are fully available without restriction, and that all data are within the manuscript. However, data reported in the manuscript are in aggregated form. My understanding is that PlosOne requires the underlying data to be made publicly available. Can the authors please clarify if the data underlying figures will be made available on a repository or as supporting information files?

Reviewer #2: Review PONE-D-20-37282 Scanning activity of elite football players in 11 v 11 match play: An eye-tracking analysis on the duration and visual information of scanning

The study presents original research on scanning information and duration as measured using eye movement registration technology. The findings largely replicate (and confirm) what has been found in other studies that used either other technology or other, less representative, task settings. The data collection and a subset of the data presented has also been published elsewhere. That is, the authors report the scanning information and duration as measured using eye movement registration technology whilst elsewhere the authors have reported the fixations as measured from the same athletes in (presumably) the same data collection. It could be questioned why the authors decided not to publish the combined dataset in a single paper.

The introduction is informative in terms of its explanation of the theoretical background and the practical relevance of investigating scanning/exploration in football. Whilst it is outlined that the study is exploratory in nature, given the existing literature that the study builds on, it would have been useful to develop and present some testable hypotheses.

The methods presented seem adequate, although data on only four players, all midfielders, is presented, which is a serious limitation in the context of the generalisability/transferability of the findings – in particular since the data are analysed/used to present group effects. Whilst acknowledging that a considerable amount of work is presented and the exploratory nature of the study, a group of N=4 are very likely not representative to all midfielders of a certain playing level. Importantly, no individual differences are discussed or interpreted in terms of their effects on the group effects. Throughout the discussion, it is important that the authors emphasise this limitation. It is not clear why one player was excluded and/or why the added inclusion criterion of having to have played in the starting line-up is relevant in the context of the aims of the study. That is, it doesn’t make sense why you would exclude 20% of your dataset on this criterion, which is unrelated to the project aims. This made me wonder whether there were other, data-integrity, reasons that may be relevant/important to report?

With the above in mind, the innovative nature of the study may need to be better focussed on the method (using eye-tracking in representative settings) presented. Importantly, the authors need to improve how they provide the information for adequate reproduction of the collected data and findings presented. For instance, the authors present in Figure 1 a schematic image detailing how the eye-tracking technology was affixed to the player. Anyone who has ever done eye-tracking in fast movement, 360-degree, contact sports knows that a lose cable (as depicted in the image) will not work. So an actual image of a player fitted with the technology would really help explain how to collect data under the circumstances of real play.

The authors need to better explain how scan initiation (and also the end/stop-point of the scan, related to scan duration) was determined. Was scan initiation purely based on head movement as determined by the movement of the video-image – or was it determined by the eye-movement in the image as determined by the eye-tracking technology? Or a combination? As a scan/exploration is a whole-body action (eye in head on body) it can be initiated by either of those (eye, head, and body). This information is important for reproduction of the findings but may also provide further important detail. If the scan initiation was determined on a combination of eye and head movement, statistics on how many of the scans were initiated by eye movements (as determined by the eye-tracker) versus head movements may provide useful further insights into the type of movement underlying the scan/exploration (see also Van Andel et al. 2019 – see below for full reference).

The explanation of variables measured needs to be more accurate and more precise. On page 10, the authors explain the analysis of two dependent variables, playing phase and player-to-ball distance, where they mean to say independent variables. To be clear, information and duration are the main outcome - dependent - variables whilst playing phase and player-to-ball distance are two independent variables, assumed/hypothesised to have an effect on the dependent variable. The same error is made later on this page when additional variables are introduced (control/pass; air/pitch; ball action). These are all independent variables. Finally, fixations during a scan (this variable also needs to be better defined), is a dependent variable.

The authors present to have analysed a total of 869 scans. How many per player? Were there individual differences between the amount of scans analysed per player that can be interpreted in terms of the methods employed (amount of time measured per player, etc.) and which can/needs to be improved on in future research? How many scans were partially or not accurately captured and or not used for further analysis? Was the total amount of scans analysed representative for the total amount of scans made by these players when they were wearing the aye-tracking technology? All of this is very important information of improvement on the method, reproduction of the findings and generalisability/transfer of the findings.

Similarly, in the results section, when the dependent vars of duration, information and scanning fixations (fixations during a scan) are discussed, it would be useful to see the number of scans per category/independent variable. Again, this would allow the reader to further understand the power of the statistics presented and related the strength of the argument and again relates to the reproducibility and generalisability of the findings.

I was a bit confused by the use of centi-seconds (tenths of a second). Why didn’t the authors use seconds as the base dimension? Related – in Figure 3 there seems to be an artefact present of using cs as a base dimension. It seems odd that the number of scans is markedly smaller for scans with a duration of 4, 8, 12, 16, etc. cs. compared to their neighbouring (2, 6, 10, 14, 18, etc. cs. scans. This is not interpreted by the authors and, as said, very likely an artefact.

The statistics presented in the results should all be announced in the methods section. That is, in the results section, the authors present quite a few ANOVA’s that are not announced in terms of their main dependent/independent measures. I would strongly recommend that the authors carefully outline all of their intended analyses in the methods section. The independent variable ‘number of players’, which apparently has two levels, is not explained anywhere (I reconstructed that the two levels were: teammate or opponent – but in Figures 7, 8 and 9 Number of players is graphed as a continuous variable – very confusing!). The authors present the ANOVA’s with repeated measures on the latter (not sure which) but it’s not clear how their observational design allows for such an analysis.

The authors should refrain from making interpretations of their data in terms of what the players actually did or did not see. For instance, on page 15 the authors describe that in the movement phases of the scans, the players most often saw zero teammates and opponents. Similarly, later on the authors describe how opponents were more visible … . There are many occasions where the authors equate having measured something being in the video-frame with it being visible (or not) or being seen (or not) by the players. To be clear, first, what is established using the video is not necessarily the same as what was available in the full field of vision. As the authors explain, the video can access an 82-degree x 52-degree portion of the complete visual scene. Arguably, this is a small portion of the full (approximate) 180-degree x 180-degree full field of vision available to the athlete, even when wearing the Tobi eye-tracking technology. Second, something can be presented in the field of vision but not be seen by the athlete. Regardless, it is always a good idea to stick to the data (what was and wasn’t on the video frame) rather than making interpretations (what was or wasn’t seen by the players) – in particular when presenting the findings in the results section but also throughout the discussion (see my later comments).

Slightly related to the above – there is a results paragraph on page 17 in which inferences are made about attention. In light of the ecological approach presented in the introduction I have reservation related to the meaningfulness of equating foveal vision during a scan to focus of attention. Hence, I found the term attention in this paragraph slightly contentious, in particular since the authors did not further explain how foveal vision may or may not relate to attention during a scan/exploration (eye-head-body movement).

The discussion contains some interesting and very relevant reflections on the data presented. In terms of the authors discussion on the scan stop and turning points being the most important part of the scan, I can recommend them to consult Van Andel et al. (2019) who may provide further detail to the different purposes of the scans in terms of their importance for orientation (where on the pitch am I in relation to other players and the ball? – relevant early on and before gaining possession of the ball) versus specification of action (what am I going to do now I have the ball, who will I pass to?).

The practical applications were quite speculative and many of the suggestions for coaches could not be derived from the data presented. Especially for the practical applications it is important that the authors stick to the evidence and make suggestions that are evidence based – and clearly linked to the evidence presented in the paper. As argued – the innovation presented is the method – as many of the findings have also been presented elsewhere, and the authors have to be careful not to overinterpret the group data presented as being representative based on the four participants and the limited amount of time per participant.

The limitations of the study should be presented before the practical implications – this should also assist the reader understand the limited generalisability based on the small participant base.

In the Conclusion section, the authors describe how the length of scanning is somewhat automated. I don’t understand this statement – what does that mean in the context of their theoretical framework (ecological approach) but importantly, on what evidence? Was there no variability? As explained earlier, in the conclusion there are many references to what players may or may not see – I wonder whether those statements are granted (or even meaningful), both based upon the data presented as well as on theoretical grounds.

van Andel, S., McGuckian, T. B., Chalkley, D., Cole, M. H., & Pepping, G.-J. (2019). Principles of the Guidance of Exploration for Orientation and Specification of Action. Frontiers in Behavioral Neuroscience, 13, 734–11. http://doi.org/10.3389/fnbeh.2019.00231

6. PLOS authors have the option to publish the peer review history of their article (what does this mean?). If published, this will include your full peer review and any attached files.

Reviewer #1: **Yes: **Dr Thomas McGuckian

Reviewer #2: No

---

## [Author Response · Author response to Decision Letter 0]

26 Feb 2021

Journal Requirements

1) We changed the manuscript according to the two attached pdf-files, as requested.

2) We have replaced the figure in question.

3) a) We have no reservations regarding providing the data as a supplementary file.

b) We added the data as a supporting supplementary file.

4) Figure 2 is not an image of a participant in the study. It is a general visualization of a scan portrayed by the study’s 2nd author. With this in mind, do you still require a consent form and amendments to the methods section? If so, we will do so promptly.

Reviewer 1

3) - On line 66, the authors report that Aksum et al. found fixation durations to be shorter in the field than in the laboratory. Given this, I am interested to know why the authors chose a 120ms duration (L240) to define a fixation in the current study, and if this may have contributed to the low number of fixations detected during scans. 

Great question! Initially, we used a 60 ms threshold for fixations. However, in the review process of our recently published article (What Do Football Players Look at? An Eye-Tracking Analysis of the Visual Fixations of Players in 11 v 11 Elite Football Match Play (Aksum et al, 2020)) we had to remove the analysis of 3388 fixations because they lasted between 60 and 120 ms (this is reported in the paper). In that case, both reviewers were adamant that any duration less than a 120 ms threshold could not be supported by the literature because all previously published studies in these types of sport contexts have operated with an approximate 120 ms threshold. Thus, we complied with these wishes. Consequently, we used the same threshold in the current paper. If, we instead use a 60 ms threshold, 138 mores scans will include fixations (approximately 18% in total). To stick with consensus in the literature, we have adhered to the 120 ms threshold overall, to enable comparisons with other studies, but because the above-mentioned finding about the 60 ms threshold is an interesting one with some potential implications, we have added the information in the supplementary data file. We are of course happy to add these 60 ms results in the text as well, together with the current results, if you feel it is warranted. 

- L127-130, while I understand this rationale, it suggests that scanning is not essential for performance for other playing positions, which seems like a difficult position to justify. Would it be appropriate to also say that central midfield players were selected as it is more often relevant for them to scan the 360-degree environment, therefore increasing the available dataset? 

Thank you for addressing this. To further the rationale of investigating central midfielders, we added a sentence with reference to Jordet et al. (2020), showing that central midfield players had higher scan frequencies than any other playing position. We feel that the word surrounded explains the 360-degree reality the midfield players perform in.

- L179-241, for clarity, I would encourage the authors to define the variables of interest in a table or to demarcate the variables with subheadings. Further, please clearly define all variables (e.g. L239, “control, no touch”, is not clear what this could mean). It would also be useful to clearly outline which are independent and dependent variables to assist with comprehending the statistical analyses and results. 

Thank you for this input. As your suggestions, we moved the variables so that we first present the dependent variables and then the different independent variables. We also made additional subheadings to provide further clarity. We have also provided further explanation on the control, not touch variable.

- Figure 3, are the authors able to provide some insight into why some durations (e.g. 26, 30, 34, etc.) occur much more often than other durations (e.g. 28, 32, 36, etc.)? It could be useful to provide this explanation within the text of the article.

Thank you for addressing this. There is a natural explanation of this result. The HD camera from the glasses filmed at 25 frames per seconds. While the overview camera filmed at 50 fps. By synchronizing those video files and analyzing them frame by frame using Assimilate Scratch Play (set at 50 fps) we were able to register scans with a 2 cs interval. However, since the camera from the glasses only produced videos of 25 fps (4 cs interval), every second frame would be blurry. In those instances where we were not certain (always odd number frames) we instructed the analysts to only register the end of a scan if the ball was certainly inside the video frame. That is why there are much more scans ending on an even number frame than an odd number frame. We have added this information in the data analyses section.

- Figure 4 caption, please indicate what the error bars are a measure of (SD, SE, etc.?). Please also do this for other figures where relevant. 

Thank you for this input. The bars always represented standard errors. Thus, we changed the wording in each caption.

- L310-313, the authors here indicate a significant difference, but earlier state that it was not found to be relevant in providing context (L229-231). Can the authors please clarify this?

As stated on line 229-231, we used three additional independent (we noted our spelling error) variables to provide context on scanning duration. One of these (air or pitch) is what was measured on line 310-313.

- L327-331, for clarity, please do not indicate that there was a difference between defence and attack. Instead, the phrasing used in lines 324-326 would more clearly outline the non-significant result. 

Thank you for this input! Revised as suggested.

- Figure 6, at first it seemed odd to me that the movement phase would have a higher number of scans with zero or one teammate/opponent than at the stop point. My reasoning was that the movement phase must include at least the number of players as the stop point, because the movement phase includes either side of the stop point. I then re-read the definition of the movement phase and noticed that the movement phase excludes players that are counted in the stop point variable (L206-207). Can the authors please provide a rationale for this? Operationally, this does not make sense to me. Teammates/opponents that are visible during the movement phase do not become invisible because the player stops moving their head, so it seems logical to include these players in both the movement phase and stop point phase variable. This would mean that for each scan, the movement phase should have at least the number of teammates/opponents as the stop point, but I do not see this as an issue. Following the definition of the movement variable (i.e. stop point players excluded), I would think that the players that are included in the foveal circle variable should be excluded from the stop point variable (but to be clear, I don’t think this would make sense either). I apologize if I am missing something clearly obvious with this comment, I will be happy to hear a good reason for this decision by the authors. 

Thank you for addressing this. We had several discussions on how to best present this data. First, we did as you commented and included all the players in all the phases. Then, we believed it was better to distinguish between players that were ONLY detected in the movement phases and the players in the stop point of the entire video frame as it would provide the cleanest results between what is possibly detected when moving from the ball to the point of interest (the stop point) and the point of interest itself. However, if the reviewer feel strongly about this we do not mind including the overlapping numbers to the movement phases as this would not change the results in any way, it would just add players from the stop point as well and make any comparison between the groups difficult.

- L357-360, I feel this extra information about the analysis would be better suited in the statistical analysis section. In doing so, please also include the level names in brackets – e.g. playing phase (2 levels: attack, defence). 

We included the level names in brackets, as suggested. We also moved it to the statistical analyses section.

- L387-389, this text is repeating what was stated earlier (L357-360), so can safely be deleted. Otherwise, it suggests that a separate analysis was being conducted, which I don’t believe is that case. 

You are correct. Removed as suggested.

4) - L407-408, d) maybe, but this is dependent on how these different parts are defined in the study (as above). For the player, all the visual information in the foveal circle are detectable in the stop point, and all the visual information in the stop point are detectable in the movement phase. 

As you correctly stated, all the visual information in the foveal circle is also a part of the stop point measure. We did it this way to examine if it was a difference in what players are (probably) focusing on when they stop their scan (foveal) and the possible information in the peripheral vision during the same “picture”. However, this is not the case between the stop point and the movement phases where it depends on the excursion of the scan. If the scan had a small excursion then all the players found in the stop point was also found in the movement phase and was then removed from the movement phase in order to not get a data overlap. In contrast, if the excursion was longer, then some players other than those in the stop point would appear in the movement phase.

- L421-431, I’m not sure I follow the logic or relationship in this argument. Can the authors more strongly put forward their point in this paragraph? Yes the ball path is more predictable when it is being passed between players, but there was a difference between passes on the ground and the air, why could this be? Yes, longer scans are operationally similar to larger excursion scans, but how does the referenced literature relate to the statements from L426 onwards? 

Thank you for addressing this. We altered the first sentence to explain that although both passes on the ground and the air have predictable paths, we believe the passes in the air has much more predictable paths because there is no opponent to possibly intercept those passes. In regard to your comments on L426-431, we agree with you that these statements are not founded in the literature. It is our understanding that no previous studies have examined how the action undertaken on the ball in the exact moment a scan is starting influence the duration of the scan. Thus, we added even more caution to our arguments by adding the following sentences: However, no previous studies have examined durations of scans or head turns in relation to the action undertaken with the ball. Hence, these assumptions should be cautiously interpreted.

- L468-478, this is a fair point and I agree with the conclusion, however, I do also again wonder about the influence of how scans are defined and the influence this might have on the outcome. For example, a scan does not start until the ball is out of the video frame (according to the definition), however the head movement before the ball is out of the frame is still part of a scanning movement. If this were included in the definition, the duration of scans would be longer. I’m not necessarily suggesting the definition of scans needs to be changed, but perhaps this is worth a mention somewhere? 

Thank you for this comment. We had to make a choice where to start examining the scans so that this measure would be 100% objective. That is why we chose to only include scans if the ball was out of the video frame and that is also the point in time when we started measuring the duration. As you correctly stated, this means that most scans will have a few more centiseconds in duration because the movement starts before the ball leaves the visual scene camera. However, this would be almost impossible to measure accurately. Additionally, this means that some micro scans will not be included because the ball never left the visual scene camera. Per your suggestion, we added more information on this in the methods section: This operationalization was constructed to ensure maximum objectivity when measuring the start and end of a scan. The limitations of this operationalization were (1) micro scans in which the ball does not leave the video frame (these were excluded from the analysis) and (2) most scans start a few unequal numbers of centiseconds before our measurement starts.

- L500-510, this one I don’t agree with so much. A limitation with the definition is alluded to at the end of this paragraph. I will be interested to hear the authors thoughts on my previous comments related to this. 

Thank you for this comment. You are addressing a point that we have had lengthy discussions on. When doing our initial analyzes we did use overlapping data points, meaning that players that were visible both in the movement phases and stop point was added in both. In the end we decided to differentiate between players that were only visible in the movement phases and the players in the stop point, so that we made certain that we did not get any overlapping data points. We believe that this is a cleaner way to present the data. However, regarding our conclusion that the stop point is more important than the movement phase, we agree that this is perhaps an unsubstantial conclusion. That is why we express this limitation in the last sentences of this paragraph. We are open to removing this statement if you feel that this would improve the article. 

- L513-515, this should maybe be phrased as ‘possible’ rather than ‘likely’, particularly given that the authors provide two very valid reasons as to why more opponents would be detected, and neither of these reasons suggest that it is the players intention to try to perceive more opponents. 

Good point! Revised as suggested.

- L533-536, could this application be strengthened by including the findings relating to attack and defence? For example, players seem to detect different information during these phases, and this information is only created because there are opponents and teammates to be detected. Therefore, unopposed exercises should be limited. 

Great input! We added the finding on different visible information in the scans in attack and defense to strengthen this recommendation. 

- L541-549, there are potentially some other limitations that would be good to include here. I would encourage the authors to outline other limitations (e.g. the minimum duration of fixation used) in full in order to assist future investigations in overcoming any possible limitations. 

Thank you for this input. We added the proposed limitation as a fourth limitation. We agree that a lower threshold would be more applicable to this type of research, but to our knowledge there is still no empirical evidence that supports fixation measurements for shorter than 120 ms because of the saccadic suppression that spills over from saccades (e.g, Discombe & Cotterill, 2015). With our current knowledge, we cannot be sure that a fixation of for example 60-100 ms in fact is a fixation and not just saccadic suppression. 

- L550-560, I agree that these are good ideas. I wonder if the authors could expand a little on why these should be future investigations. Are they particularly under investigated? Will they bring about important practical applications? Will they answer questions that came up in the current study? Thank you for this input! We added a few sentences explaining that the method used in this study should be used in future research, as it has never been used before in this setting. We also added a sentence on practical implications, and encouraged researchers to answer the questions that arise from our findings.

5) - L84-85, I feel the wording of this sentence gives the impression that affordances are only relevant for on-ball activity. This is not correct from the broader theoretical standpoint, but also in this paper which investigates both attack and defence. To clarify this sentence, I feel it would be useful to say something like “In football, affordances involving interaction with the ball rely heavily…”. 

Great input! We emphasized that affordances occur in both attack and defense and provided an attacking example of exploration before receiving the ball using your wording.

- L88-90, I feel the authors could be more direct with the explanation of the ecological approach here. Under an ecological approach, perception-action couplings are context specific (not implied), and therefore they need to be studied in the performance environment to be explained. 

Thank you for this input. We changed the sentence to: Furthermore, it greatly informed our research design because, according to the ecological approach, perception-action couplings are context-specific and have to be studied in the performance environment that the research aims to explain

- L91, I’m not sure this sentence is very clear. Would it suffice to just say that “Visual scanning has been analysed…”? 

Great input. Revised as suggested.

- L98, should this be scanning frequently, not frequency? If the former is used, this will explain the relationship (positive effect). With the latter, it is not clear (at least, it could be more clear) if it is a high or low frequency that has the positive effect on passing performance. 

Thank for noticing this. We changed it to higher scan frequency.

- L429, ‘tits’, a small typo, I’m assuming. 

A small one, yes!

7) - The authors report that data are fully available without restriction, and that all data are within the manuscript. However, data reported in the manuscript are in aggregated form. My understanding is that PlosOne requires the underlying data to be made publicly available. Can the authors please clarify if the data underlying figures will be made available on a repository or as supporting information files? 

We were not aware of this requirement when we first submitted the article. The data has now been submitted in a supporting SPSS file.

Reviewer 2

The study presents original research on scanning information and duration as measured using eye movement registration technology. The findings largely replicate (and confirm) what has been found in other studies that used either other technology or other, less representative, task settings. The data collection and a subset of the data presented has also been published elsewhere. That is, the authors report the scanning information and duration as measured using eye movement registration technology whilst elsewhere the authors have reported the fixations as measured from the same athletes in (presumably) the same data collection. It could be questioned why the authors decided not to publish the combined dataset in a single paper. 

Thank you for your comments. Regarding the fixations – A very small amount of our results (20 out of 2832 fixations) were analyzed in a previous article (What Do Football Players Look at? An Eye-Tracking Analysis of the Visual Fixations of Players in 11 v 11 Elite Football Match Play (Aksum et al, 2020)). These registrations are presented in the current article as an independent variable for one single analysis. These registrations represent a secondary and negligible part of our results. Despite this minor overlap with that previous paper, we believe it adds context, value, and completeness to the current paper and we have chosen to keep those registrations. Regarding the choice to publish this data in two separate papers, the reason is that these are two entirely different studies, building on very different research traditions. The first paper examines visual fixations, and belongs in the laboratory based, visual fixation tradition (although we took the study to the field). Whereas the current paper examines scanning, and belongs in the field-based visual exploratory activity tradition. We strongly believe that combining those two research questions and traditions in the same paper would make the paper very confusing to both read and interpret. 

The introduction is informative in terms of its explanation of the theoretical background and the practical relevance of investigating scanning/exploration in football. Whilst it is outlined that the study is exploratory in nature, given the existing literature that the study builds on, it would have been useful to develop and present some testable hypotheses. 

Thank you for this feedback. Although some studies of scanning exist, no studies (to our knowledge) have ever examined the duration and visible information that players look at during scanning. Thus, we believe that the basis for making hypotheses were not there. Hopefully, hypotheses could be created based on the current study’s results.

The methods presented seem adequate, although data on only four players, all midfielders, is presented, which is a serious limitation in the context of the generalizability/transferability of the findings – in particular since the data are analysed/used to present group effects. Whilst acknowledging that a considerable amount of work is presented and the exploratory nature of the study, a group of N=4 are very likely not representative to all midfielders of a certain playing level. Importantly, no individual differences are discussed or interpreted in terms of their effects on the group effects. Throughout the discussion, it is important that the authors emphasize this limitation. It is not clear why one player was excluded and/or why the added inclusion criterion of having to have played in the starting line-up is relevant in the context of the aims of the study. That is, it doesn’t make sense why you would exclude 20% of your dataset on this criterion, which is unrelated to the project aims. This made me wonder whether there were other, data-integrity, reasons that may be relevant/important to report? 

We agree that the number of players in the same position is a limitation. Thus, we have highlighted this limitation early in the ‘discussion’ and ‘limitation’ sections. Regarding the choice to exclude one of the participants, this was done to keep this study consistent with the samples in some of the previous scanning studies (e.g., Jordet, 2015; Jordet et al., 2020), which exclusively consisted of professional, first team players. Although this was the aim from the outset when we set out to collect data in a professional first team environment, we realized after the data was collected that one of the players had no first team games, and that including him would compromise the otherwise homogenous elite group. Consequently, we decided to include only those four players. Also, importantly, the decision to exclude this player was made without any knowledge about the results this player would add, as the decision was made prior to any data was analyzed. We have elaborated on this point in the participants section of the revised manuscript. We also added some individual numbers of scans and fixations, as suggested.

With the above in mind, the innovative nature of the study may need to be better focused on the method (using eye-tracking in representative settings) presented. Importantly, the authors need to improve how they provide the information for adequate reproduction of the collected data and findings presented. For instance, the authors present in Figure 1 a schematic image detailing how the eye-tracking technology was affixed to the player. Anyone who has ever done eye-tracking in fast movement, 360-degree, contact sports knows that a lose cable (as depicted in the image) will not work. So an actual image of a player fitted with the technology would really help explain how to collect data under the circumstances of real play. 

Thank you for these comments. We decided to change Figure 1 to a figure of one of the participants wearing the technology (below). This image shows how we attached the eye-tracker battery on the upper back of the players, allowing them to have maximum mobility and freedom without the fear of detaching the battery. The players also reported that they were able to play without any restrictions. We only expressed to them that we would appreciate if they did not head the ball. 

The authors need to better explain how scan initiation (and also the end/stop-point of the scan, related to scan duration) was determined. Was scan initiation purely based on head movement as determined by the movement of the video-image – or was it determined by the eye-movement in the image as determined by the eye-tracking technology? Or a combination? As a scan/exploration is a whole-body action (eye in head on body) it can be initiated by either of those (eye, head, and body). This information is important for reproduction of the findings but may also provide further important detail. If the scan initiation was determined on a combination of eye and head movement, statistics on how many of the scans were initiated by eye movements (as determined by the eye-tracker) versus head movements may provide useful further insights into the type of movement underlying the scan/exploration (see also Van Andel et al. 2019 – see below for full reference). 

Great questions! As stated in the manuscript, both the initiation of the scan and stop point (turning point) of the scan was determined by the movement of the head (the ball went out of the visual scene camera). It was operationalized in this way to ensure maximum objectivity and reproductivity. We added some information to hopefully make this clearer in the revised manuscript. Thank you for making us aware of that article. Most scans in this article would be understood as exploration for orientation because we do not measure scans when a player is in possession of the ball (according to earlier definitions (Jordet, 2005, Jordet et al. 2020). This information has now been added to the revised manuscript in the variables section of the Methods. However, scans conducted before receiving the ball would be more related to exploration for action specification. There was no such variable included in this study. We added a reference to the van Andel article in the ‘future research’ section.

The explanation of variables measured needs to be more accurate and more precise. On page 10, the authors explain the analysis of two dependent variables, playing phase and player-to-ball distance, where they mean to say independent variables. To be clear, information and duration are the main outcome - dependent - variables whilst playing phase and player-to-ball distance are two independent variables, assumed/hypothesized to have an effect on the dependent variable. The same error is made later on this page when additional variables are introduced (control/pass; air/pitch; ball action). These are all independent variables. Finally, fixations during a scan (this variable also needs to be better defined), is a dependent variable. 

Thank you for making us aware of this. Frankly, these are embarrassing mistakes to make. We revised the variables you mentioned. We also added a more comprehensive explanation of fixation detection which we changed to “the presence of fixations” in the revised manuscript. Regarding your last point, fixations are not used as a dependent variable, it is used an independent variable on the dependent variable scanning duration. We changed the order of the presented variables to enhance readability and improve the logical sequence.

The authors present to have analysed a total of 869 scans. How many per player? 

The individual number of scans were: Player 1 = 381, Player 2 = 208, Player 3 = 177, Player 4 = 103. We added this in the variables section of the revised manuscript. Were there individual differences between the amount of scans analysed per player that can be interpreted in terms of the methods employed (amount of time measured per player, etc.) and which can/needs to be improved on in future research? Yes. As mentioned in the methods section, two players wore the glasses for 10 minutes each and two players wore them for 20 minutes each. Furthermore, the nature of an actual football match means that we cannot control for how much time the ball is in play and how much time the ball is out of play (i.e., free-kicks, corners). That is why we included data from all players irrespective of duration and that is also why it was difficult to compare results between the players. 

How many scans were partially or not accurately captured and or not used for further analysis? 

All scans conducted by the four players were captured and analyzed. Was the total amount of scans analysed representative for the total amount of scans made by these players when they were wearing the aye-tracking technology? All scans conducted by the four players were captured and analyzed. All of this is very important information of improvement on the method, reproduction of the findings and generalisability/transfer of the findings. 

We agree. We added information specifying that all scans were analyzed and the individual number of scans from each player. 

Similarly, in the results section, when the dependent vars of duration, information and scanning fixations (fixations during a scan) are discussed, it would be useful to see the number of scans per category/independent variable. Again, this would allow the reader to further understand the power of the statistics presented and related the strength of the argument and again relates to the reproducibility and generalizability of the findings. 

Thank you for addressing this. Although, we were a bit uncertain of what you are requesting at this particular point, we have tried and added the individual numbers of scans in the methods section. We also added individual number of fixations in scanning. 

I was a bit confused by the use of centi-seconds (tenths of a second). Why didn’t the authors use seconds as the base dimension? Related – in Figure 3 there seems to be an artefact present of using cs as a base dimension. It seems odd that the number of scans is markedly smaller for scans with a duration of 4, 8, 12, 16, etc. cs. compared to their neighboring (2, 6, 10, 14, 18, etc. cs. scans. This is not interpreted by the authors and, as said, very likely an artefact. 

Thank you for addressing this. Centiseconds (one hundredth of one second) was used because we believe it is best suited to display our data material. As the videos was produced at 50 and 25 fps we would get results as accurately as 2 centiseconds (0.02 seconds) for each scan. The artefact of Figure 3 has nothing to do with the use of cs, but has to do with the video from the HD-camera on the eye tracker glasses. Meaning that, although we analyzed data at 50 fps, every second frame would sometimes be blurry (as they are a combination of two frames), which meant that it was hard to detect whether the ball was back inside the video frame or not. We instructed the analysts to be certain that the ball had returned in the video frame before they registered the end of a scan. That is why there are more registrations on odd frame numbers than even frame numbers. We have provided more information on this in the data analyses section. We also added in the first sentence of the results section the number 39.65 cs in seconds (0.3965 s) to show how noisy it would be to present accurate data in seconds. 

The statistics presented in the results should all be announced in the methods section. That is, in the results section, the authors present quite a few ANOVA’s that are not announced in terms of their main dependent/independent measures. I would strongly recommend that the authors carefully outline all of their intended analyses in the methods section. The independent variable ‘number of players’, which apparently has two levels, is not explained anywhere (I reconstructed that the two levels were: teammate or opponent – but in Figures 7, 8 and 9 Number of players is graphed as a continuous variable – very confusing!). The authors present the ANOVA’s with repeated measures on the latter (not sure which) but it’s not clear how their observational design allows for such an analysis. 

We agree! Thanks you for commenting this. As requested, we have rewritten the paragraph of the statistical analysis in the method section so that the intended analyses are outlined in a clearer manner. With respect to the dependent variable ‘number of players’, we do think that it is explained well, however we conducted some minor changes to this. Additionally, the three-way ANOVAs with scanning information as repeated measures allowed us to compare number of teammates and opponents as a potential factor of information sources during a scan.

The authors should refrain from making interpretations of their data in terms of what the players actually did or did not see. For instance, on page 15 the authors describe that in the movement phases of the scans, the players most often saw zero teammates and opponents. Similarly, later on the authors describe how opponents were more visible … . There are many occasions where the authors equate having measured something being in the video-frame with it being visible (or not) or being seen (or not) by the players. To be clear, first, what is established using the video is not necessarily the same as what was available in the full field of vision. As the authors explain, the video can access an 82-degree x 52-degree portion of the complete visual scene. Arguably, this is a small portion of the full (approximate) 180-degree x 180-degree full field of vision available to the athlete, even when wearing the Tobi eye-tracking technology. Second, something can be presented in the field of vision but not be seen by the athlete. Regardless, it is always a good idea to stick to the data (what was and wasn’t on the video frame) rather than making interpretations (what was or wasn’t seen by the players) – in particular when presenting the findings in the results section but also throughout the discussion (see my later comments). 

Thank you for this input! You are absolutely correct, and we have changed all the terminology related to seeing and visibility. This has greatly improved our results and discussion section.

Slightly related to the above – there is a results paragraph on page 17 in which inferences are made about attention. In light of the ecological approach presented in the introduction I have reservation related to the meaningfulness of equating foveal vision during a scan to focus of attention. Hence, I found the term attention in this paragraph slightly contentious, in particular since the authors did not further explain how foveal vision may or may not relate to attention during a scan/exploration (eye-head-body movement).

In the discussion we use the reference (Nakashima R, Shioiri S. Why Do We Move Our Head to Look at an Object in Our Peripheral Region? Lateral Viewing Interferes with Attentive Search. PLOS ONE. 2014;9(3):e92284. doi: 10.1371/journal.pone.0092284) to explain that the foveal eye position is often similar to or the equivalent of attention. Thus, we believe that naming it attention can be substantiated by others empirical research. We also see no problems of using the word attention in relation to ecological theories on visual perception. However, we agree with you that attention is not always the same as the foveal gaze point. Hence, we removed the wording completely from the paragraph in the results section.

The discussion contains some interesting and very relevant reflections on the data presented. In terms of the authors discussion on the scan stop and turning points being the most important part of the scan, I can recommend them to consult Van Andel et al. (2019) who may provide further detail to the different purposes of the scans in terms of their importance for orientation (where on the pitch am I in relation to other players and the ball? – relevant early on and before gaining possession of the ball) versus specification of action (what am I going to do now I have the ball, who will I pass to?). 

Thank you for this suggestion. The work of van Andel and colleagues are very interesting and we will surely consult them in the future. We added a reference to their work in the future research section. For this article, as we do not measure scans when a player has possession of the ball (Jordet, 2005, Jordet et al, 2020) we found it difficult to discuss the results using the exploration for action specification and exploration for orientation difference as almost all of the scans would be categorized as exploration for orientation.

The practical applications were quite speculative and many of the suggestions for coaches could not be derived from the data presented. Especially for the practical applications it is important that the authors stick to the evidence and make suggestions that are evidence based – and clearly linked to the evidence presented in the paper. As argued – the innovation presented is the method – as many of the findings have also been presented elsewhere, and the authors have to be careful not to overinterpret the group data presented as being representative based on the four participants and the limited amount of time per participant. 

Thank you for these comments. We removed the last paragraph as it did not have a direct link to our results. We also added a sentence at the bottom expressing extreme caution when interpreting the results. Additionally, we added cautious words to the practical applications such as ‘likely’ and ‘probably’.

The limitations of the study should be presented before the practical implications – this should also assist the reader understand the limited generalizability based on the small participant base. 

Revised as suggested.

In the Conclusion section, the authors describe how the length of scanning is somewhat automated. I don’t understand this statement – what does that mean in the context of their theoretical framework (ecological approach) but importantly, on what evidence? Was there no variability? As explained earlier, in the conclusion there are many references to what players may or may not see – I wonder whether those statements are granted (or even meaningful), both based upon the data presented as well as on theoretical grounds. 

Thank you for making us aware of this. We have removed the statements of automation in both the discussion and conclusion. We have also removed every statement regarding what players saw or did not see.

---

## [Decision Letter · Decision Letter 1]

14 May 2021

PONE-D-20-37282R1

Scanning activity of elite football players in 11 vs. 11 match play: An eye-tracking analysis on the duration and visual information of scanning

PLOS ONE

Dear Dr. Aksum,

Thank you for submitting your manuscript to PLOS ONE. After careful consideration, we feel that it has merit but does not fully meet PLOS ONE’s publication criteria as it currently stands. Therefore, we invite you to submit a revised version of the manuscript that addresses the points raised during the review process. You will see that these are minor revisions.

We look forward to receiving your revised manuscript.

Kind regards,

Greg Wood, PhD

Academic Editor

PLOS ONE

Journal Requirements:

Reviewers' comments:

Reviewer's Responses to Questions

**Comments to the Author**

1. If the authors have adequately addressed your comments raised in a previous round of review and you feel that this manuscript is now acceptable for publication, you may indicate that here to bypass the “Comments to the Author” section, enter your conflict of interest statement in the “Confidential to Editor” section, and submit your "Accept" recommendation.

Reviewer #1: (No Response)

Reviewer #2: (No Response)

2. Is the manuscript technically sound, and do the data support the conclusions?

Reviewer #1: Partly

Reviewer #2: Yes

3. Has the statistical analysis been performed appropriately and rigorously? 

Reviewer #1: Yes

Reviewer #2: Yes

4. Have the authors made all data underlying the findings in their manuscript fully available?

Reviewer #1: Yes

Reviewer #2: Yes

5. Is the manuscript presented in an intelligible fashion and written in standard English?

Reviewer #1: Yes

Reviewer #2: Yes

6. Review Comments to the Author

Reviewer #1: I am happy to see the revisions made in the current version of the manuscript. I believe the paper is already much improved in its clarity, and the paper makes good logical sense. I have some concerns still (as outlined below), however I feel everything else in the manuscript is now much more strongly communicated. Well done so far to the authors.

- Overall, my comments below mostly relate to the operational definition of scanning information between movement and stop-point phases. I feel the current definition causes issues in interpretation of the findings, and often results in statements that could easily be falsely interpreted. The authors have offered to change their definition for movement phases to include information that is also included in the stop point. I will leave this to the authors to decide, however if they do wish to keep the current definition, they should be very careful about how the findings are interpreted and discussed.

- L240-241, can the authors please clarify this statement that these independent variables were not found to be relevant in providing further context for scanning information? It seems that this should be presented as a finding, rather than in the definition of independent variables. If a lack of difference in scanning information for these variables was found, this is still a finding, and should not be interpreted as providing a lack of context.

- L381-383, the players most often had zero players in the video frame during movement phases (Fig. 6). This is due to the categorisation of information to movement and stop point phases, right? If so, this statement should be clarified so as not to cause confusion for a reader.

- L391-393, I don’t know that the authors can state this, as this is clearly an artifact of the operational definition used in this study. I am fine with the authors using this definition, but it is my opinion that this type of summary statement cannot be used.

- L444-445, for (b) for the same reason as (d) below, I don’t think this claim can be made with confidence. This is surely due to the definition used in the current study, and is not representative of the visual information that actually is available during both the movement and stop point phases? The separation of these variables is fine, but I do not believe that this sort of discussion (L544-554) can be made, knowing that it is entirely due to the definition and data analysis used.

- L446-447, for (d), this is only true because of the definition used in this study and the way data has been categorised. Again, there is information overlap between aspects of a scan, which is being ignored by the operational definition used in this case. In fact, if the definition would be reversed, such that the stop point phase did not include teammates and opponents that were detected in the movement phase, this finding would likely be opposite. Therefore, this sort of statement needs to be made carefully to ensure the actual finding is not overlooked. Rather than rationalise with the acceptance of the limitations, it would be better to not make these claims in the first place. Please amend.

- L512-522, this is an interesting finding that is in line with previous research. However, it should be noted that part of this may again be due to the method of categorising scanning actions to start when the ball leaves the video frame. It is worth noting that the differences in classifying visual scanning actions between study methodologies, and the potential influence of this on findings.

Reviewer #2: I want to commend the authors on their revision of the manuscript. They have done a great job revising the manuscript and have addressed most, if not all (apart from what is indicated below) of my previous comments and reservations.

My main concern (and associated advice) is that there are still occasions where the authors make statements that cannot be supported by the data collected and theory and analyses presented. For example, in the abstract, the authors conclude that the players detected more opponents than teammates (line 37). As also argued before, it is impossible with the presented methodology to establish what the participants actually saw or detected. Again, the authors equate having measured something being in the video-frame with it being detected (or not) or being seen (or not) by the players. What is established using the video is not necessarily the same as what was available in the full field of vision and something can be presented in the field of vision but not be seen by the athlete. So, my advice again is for the authors to stick to the data (what was and wasn’t on the video frame) rather than making interpretations (what was or wasn’t seen or detected by the players) – in particular when presenting the findings in the abstract (line 37) but also throughout, in the results, discussion (specifically, line 501/502; line 524-525 and paragraph from line 523) and conclusion (line 629).

Further, more detailed comments:

Line 239-240. It is announced that four independent variables were used to provide further context for scanning duration but that they were found NOT to be relevant. In what way were they not relevant and if they were not relevant, why are they presented and discussed? I was a bit confused by this statement.

Line 243 and beyond (whole paragraph and line 443 in discussion). To assist the reader to appreciate this paragraph, further clarification is needed as earlier it is stated that no scans were analysed where the player/participant themselves were in possession. This paragraph refers to possessions, but presumably of other players – NOT the participants themselves. This needs to be made more explicit and obvious for proper understanding of the methodology and findings. Mixed usage of players (i.e. other players) versus participants (i.e. the individuals who’s scans are collected and analysed) can cause this confusion.

Line 356. As a methodological debate/discussion/argument will be developed in the discussion in regards to the ‘fixation’ cut-off of 120ms (as defined in the methods), it may be useful to emphasise throughout how fixations were determined by the current method. In the discussion reference is made to an alternative analysis with a 60ms fixation detection threshold. Authors may want to develop this argument a bit earlier as it is a very relevant finding of the current study that, based upon a fixation detection threshold of 120ms – and the mean scanning duration being 397ms. Future research will need to be aware of the need to scrutinise the currently ‘accepted’ fixation threshold of 120ms which has come about in research under lab conditions and quite often stationary (no head movements) tasks and frontal projections. Further references to literature making similar methodological arguments will be useful.

Line 450. … therefor … can be deleted.

Lines 439 and beyond. See also my comments above on definition of a fixation and presenting conclusions that are not grounded in evidence. Importantly, it is argued that fixations are absent – but there is also an argument that the fixation threshold may be too strict to pick up fixations. These two arguments partly contradict each other. Further, it is argued that the ‘absence’ of fixations evidences that they did not ‘see’ their opponents and teammates in ‘high definition’ (line 497/498); that information intake seldom (sometimes, never?) originates from ‘clear foveal images’; and that the findings merit understanding of scanning as a “blurred video”. There are quite a few unfounded or at best not discussed assumptions at play here about how human perception (and action) may function and these (non-apparent) assumptions are clearly not aligned with the explicit theoretical framework of ecological psychology that is introduced. One big assumptions from this framework is that human perception (and action) should NOT be equated (or compared!) to pictorial perception (Gibson, 1979). That is, the stimulus for visual perception is the optic array (the environment) – NOT the retinal image, so perception (and action) is NOT like watching images or videos. The explanation and understanding of visual perception as pictorial perception outlined in this paragraph is very confusing and unhelpful for proper interpretation of the findings and methodological and theoretical contribution of the paper. This paragraph, as well as the Practical applications paragraph (on lines 587/588 there is another reference to clear high definition pictures, which is problematic), need to be reworked to better align the argument to the theoretical framework of ecological psychology and to the methodological argument (and contribution) of the limitations of using a certain fixation threshold.

Lines 597 and beyond. It wasn’t apparent to me how this advice could be directly derived from the findings presented. This may have been observed in these players, but does that merit that practice should be organised accordingly? And does that transfer to any level of practice – or only practice at the highest level? What about early learners? What if we want to teach the experts something new? The findings represent real world behaviour but that is not to say that it was optimal behaviour, simply because these were high-level athletes.

7. PLOS authors have the option to publish the peer review history of their article (what does this mean?). If published, this will include your full peer review and any attached files.

Reviewer #1: No

Reviewer #2: No

---

## [Author Response · Author response to Decision Letter 1]

23 Jun 2021

Reviewer #1: I am happy to see the revisions made in the current version of the manuscript. I believe the paper is already much improved in its clarity, and the paper makes good logical sense. I have some concerns still (as outlined below), however I feel everything else in the manuscript is now much more strongly communicated. Well done so far to the authors.

- Overall, my comments below mostly relate to the operational definition of scanning information between movement and stop-point phases. I feel the current definition causes issues in interpretation of the findings, and often results in statements that could easily be falsely interpreted. The authors have offered to change their definition for movement phases to include information that is also included in the stop point. I will leave this to the authors to decide, however if they do wish to keep the current definition, they should be very careful about how the findings are interpreted and discussed. 

Thank you for this input. We decided to keep the definition as it is. However, we have made our interpretations of these findings as well as the following discussion much more cautious.

- L240-241, can the authors please clarify this statement that these independent variables were not found to be relevant in providing further context for scanning information? It seems that this should be presented as a finding, rather than in the definition of independent variables. If a lack of difference in scanning information for these variables was found, this is still a finding, and should not be interpreted as providing a lack of context.

Thank you for this input. I understand the confusion. We believed that control or pass, air or pitch, ball action, and the presence of fixations would add interesting information to the examination of scanning duration. Oppositely, we believed that playing phase and player-to-ball distance would provide enough context for the scanning information variable. There was never any plan to also include those four contextual variables for scanning duration. We decided to remove the entire sentence in order to not cause confusion.

- L381-383, the players most often had zero players in the video frame during movement phases (Fig. 6). This is due to the categorisation of information to movement and stop point phases, right? If so, this statement should be clarified so as not to cause confusion for a reader. 

Thank you for this input! We added a sentence explaining that this result should be seen in light of our operationalization of the movement phase.

- L391-393, I don’t know that the authors can state this, as this is clearly an artifact of the operational definition used in this study. I am fine with the authors using this definition, but it is my opinion that this type of summary statement cannot be used.

We fully agree with your comment. We have removed that summary statement.

- L444-445, for (b) for the same reason as (d) below, I don’t think this claim can be made with confidence. This is surely due to the definition used in the current study, and is not representative of the visual information that actually is available during both the movement and stop point phases? The separation of these variables is fine, but I do not believe that this sort of discussion (L544-554) can be made, knowing that it is entirely due to the definition and data analysis used. 

Thank you for this input. We changed the sentence from visual information to the number of teammates and opponents. We also made changes to the paragraph L544-554 according to your suggestions.

- L446-447, for (d), this is only true because of the definition used in this study and the way data has been categorised. Again, there is information overlap between aspects of a scan, which is being ignored by the operational definition used in this case. In fact, if the definition would be reversed, such that the stop point phase did not include teammates and opponents that were detected in the movement phase, this finding would likely be opposite. Therefore, this sort of statement needs to be made carefully to ensure the actual finding is not overlooked. Rather than rationalise with the acceptance of the limitations, it would be better to not make these claims in the first place. Please amend. 

Thank you for this input. We agree an added “based on our operationalizations” at the beginning of the sentence.

- L512-522, this is an interesting finding that is in line with previous research. However, it should be noted that part of this may again be due to the method of categorising scanning actions to start when the ball leaves the video frame. It is worth noting that the differences in classifying visual scanning actions between study methodologies, and the potential influence of this on findings. 

Great input! We added a sentence explaining that this result could be partly influenced by our operationalization which started measuring the duration at the moment the ball left the video frame.

Reviewer #2: I want to commend the authors on their revision of the manuscript. They have done a great job revising the manuscript and have addressed most, if not all (apart from what is indicated below) of my previous comments and reservations.

My main concern (and associated advice) is that there are still occasions where the authors make statements that cannot be supported by the data collected and theory and analyses presented. For example, in the abstract, the authors conclude that the players detected more opponents than teammates (line 37). As also argued before, it is impossible with the presented methodology to establish what the participants actually saw or detected. Again, the authors equate having measured something being in the video-frame with it being detected (or not) or being seen (or not) by the players. What is established using the video is not necessarily the same as what was available in the full field of vision and something can be presented in the field of vision but not be seen by the athlete. So, my advice again is for the authors to stick to the data (what was and wasn’t on the video frame) rather than making interpretations (what was or wasn’t seen or detected by the players) – in particular when presenting the findings in the abstract (line 37) but also throughout, in the results, discussion (specifically, line 501/502; line 524-525 and paragraph from line 523) and conclusion (line 629). 

Thank you for this input! We changed all instances where we referred to seeing or detection. We believe the manuscript has benefited greatly from this. 

Further, more detailed comments:

Line 239-240. It is announced that four independent variables were used to provide further context for scanning duration but that they were found NOT to be relevant. In what way were they not relevant and if they were not relevant, why are they presented and discussed? I was a bit confused by this statement.

Thank you for this input. I understand the confusion. We believed that control or pass, air or pitch, ball action, and the presence of fixations would add interesting information to the examination of scanning duration. Oppositely, we believed that playing phase and player-to-ball distance would provide enough context for the scanning information variable. There was never any plan to also include those four contextual variables for scanning duration. We decided to remove the entire sentence in order to not cause confusion.

Line 243 and beyond (whole paragraph and line 443 in discussion). To assist the reader to appreciate this paragraph, further clarification is needed as earlier it is stated that no scans were analysed where the player/participant themselves were in possession. This paragraph refers to possessions, but presumably of other players – NOT the participants themselves. This needs to be made more explicit and obvious for proper understanding of the methodology and findings. Mixed usage of players (i.e. other players) versus participants (i.e. the individuals who’s scans are collected and analysed) can cause this confusion. 

Great input! We added a sentence at the start of the first paragraph to emphasize this point. 

Line 356. As a methodological debate/discussion/argument will be developed in the discussion in regards to the ‘fixation’ cut-off of 120ms (as defined in the methods), it may be useful to emphasise throughout how fixations were determined by the current method. In the discussion reference is made to an alternative analysis with a 60ms fixation detection threshold. Authors may want to develop this argument a bit earlier as it is a very relevant finding of the current study that, based upon a fixation detection threshold of 120ms – and the mean scanning duration being 397ms. Future research will need to be aware of the need to scrutinise the currently ‘accepted’ fixation threshold of 120ms which has come about in research under lab conditions and quite often stationary (no head movements) tasks and frontal projections. Further references to literature making similar methodological arguments will be useful. 

Great input! We included a few sentences in the methods section in which we further explained the use of a 120 ms threshold while argued that lower thresholds, once accepted in the scientific community, might be better. Additionally, we added information that we used a 120 ms threshold when we presented the fixation results. Regarding your suggestion of adding references that supports this methodological argument, we were unable to find any. Could you please be specific regarding what paper(-s) you are referring to.

Line 450. … therefor … can be deleted. 

Revised as suggested.

Lines 439 and beyond. See also my comments above on definition of a fixation and presenting conclusions that are not grounded in evidence. Importantly, it is argued that fixations are absent – but there is also an argument that the fixation threshold may be too strict to pick up fixations. These two arguments partly contradict each other. Further, it is argued that the ‘absence’ of fixations evidences that they did not ‘see’ their opponents and teammates in ‘high definition’ (line 497/498); that information intake seldom (sometimes, never?) originates from ‘clear foveal images’; and that the findings merit understanding of scanning as a “blurred video”. There are quite a few unfounded or at best not discussed assumptions at play here about how human perception (and action) may function and these (non-apparent) assumptions are clearly not aligned with the explicit theoretical framework of ecological psychology that is introduced. One big assumptions from this framework is that human perception (and action) should NOT be equated (or compared!) to pictorial perception (Gibson, 1979). That is, the stimulus for visual perception is the optic array (the environment) – NOT the retinal image, so perception (and action) is NOT like watching images or videos. The explanation and understanding of visual perception as pictorial perception outlined in this paragraph is very confusing and unhelpful for proper interpretation of the findings and methodological and theoretical contribution of the paper. This paragraph, as well as the Practical applications paragraph (on lines 587/588 there is another reference to clear high definition pictures, which is problematic), need to be reworked to better align the argument to the theoretical framework of ecological psychology and to the methodological argument (and contribution) of the limitations of using a certain fixation threshold.

Thank you for these comments. (1) Regarding the seemingly contradictory statements of fixations, we have to present the results from the accepted 120 ms threshold. As we mentioned in our last response, adopting a 60 ms threshold would result in approximately 18% of scans with fixations. Thus, we believe that we confidently can say that fixations are rare in scans, whilst adding the information that a lower threshold should perhaps be implemented in future research. (2) We removed or changed the wording “high definition”, “clear foveal images”, and “blurred video” to better align with our theoretical framework. 

Lines 597 and beyond. It wasn’t apparent to me how this advice could be directly derived from the findings presented. This may have been observed in these players, but does that merit that practice should be organised accordingly? And does that transfer to any level of practice – or only practice at the highest level? What about early learners? What if we want to teach the experts something new? The findings represent real world behaviour but that is not to say that it was optimal behaviour, simply because these were high-level athletes. 

This is a great input! We made several alterations to the practical applications for coaches, mostly related to the wording. We included words like “in line with our results” and “inspired by our results” instead of saying that it is directly derived. We also changed the wording from should to could and so on.

---

## [Decision Letter · Decision Letter 2]

28 Jul 2021

PONE-D-20-37282R2

Scanning activity of elite football players in 11 vs. 11 match play: An eye-tracking analysis on the duration and visual information of scanning

PLOS ONE

Dear Dr. Aksum,

Thank you for submitting your manuscript to PLOS ONE. After careful consideration, we feel that it has merit but does not fully meet PLOS ONE’s publication criteria as it currently stands. Therefore, we invite you to submit a revised version of the manuscript that addresses the points raised during the review process.

There are some very minor revision proposed by a reviewer. Once these have been completed, I will accept your manuscript without further review. 

We look forward to receiving your revised manuscript.

Kind regards,

Greg Wood, PhD

Academic Editor

PLOS ONE

Journal Requirements:

Reviewers' comments:

Reviewer's Responses to Questions

**Comments to the Author**

1. If the authors have adequately addressed your comments raised in a previous round of review and you feel that this manuscript is now acceptable for publication, you may indicate that here to bypass the “Comments to the Author” section, enter your conflict of interest statement in the “Confidential to Editor” section, and submit your "Accept" recommendation.

Reviewer #1: All comments have been addressed

Reviewer #2: All comments have been addressed

2. Is the manuscript technically sound, and do the data support the conclusions?

Reviewer #1: Yes

Reviewer #2: Yes

3. Has the statistical analysis been performed appropriately and rigorously? 

Reviewer #1: Yes

Reviewer #2: Yes

4. Have the authors made all data underlying the findings in their manuscript fully available?

Reviewer #1: Yes

Reviewer #2: Yes

5. Is the manuscript presented in an intelligible fashion and written in standard English?

Reviewer #1: Yes

Reviewer #2: Yes

6. Review Comments to the Author

Reviewer #1: (No Response)

Reviewer #2: I want to commend the authors on this further revision of the manuscript – it has really improved the paper and I am happy to recommend to accept the paper pending some minor edits as per below.

Throughout the authors should check that past tense is used when relevant. Below my pick-ups:

Line 79: Jordet suggested …

Line 121: As such, we aimed …

Line 123: the aim of this exploratory study was …

Line 243: operationalizations were made …

Line 455: initiation influenced …

Line 456: dinstance influenced …

Line 457: scans involved …

Line 458: scan revealed …

Line 564: players moved …

Line 564: until they arrived …

Line 565: they wished …

Lines 263/264 (and throughout): Please refer to fixation detection threshold throughout, so insert ‘fixation detection’ between ‘lower’ and ‘threshold’ here.

Line 264: It may be a while until a lower fixation detection threshold is accepted in the context of scanning. In light of this, consider replacing ‘accepted’ with ‘considered’.

Lines 396-398: This sentence confused me. Maybe, for added clarity, add ‘in the video frame’ after ‘… more than seven teammates’ and after ‘… nine opponents’

Line 421: ‘while in defence’

Line 422/421: Consider rephrasing , eg: ‘… while in defence, no differences could be found between the amount of opponents and teammates in the video frame’.

Line 474: ‘the ball is being passed along the ground …’

Line 476: ‘the pass is made through the air …’

Line 510-513: This reads like a left-over from a previous version. The reference to clear foveal information, as outline before, makes no sense from an ecological framework, and makes an implicit reference to perception of pictures. What’s more, whereas the manuscript now contributes to the readers understanding of the role of the fixation detection threshold in the determination of fixations, this sentence weakens that contributions (as you later explain in lines 516 – 523).

Lines 652/653: As above, this sentence seems a left-over on the ‘seeing detail’ part (confusing perception and action with pictorial perception) and weakens the argument. I strongly suggest the authors rephrase or delete this very suggestive supposition that is neither based on theory nor finds any evidence in data presented.

Line 517/518: As above, write ‘lower fixation detection threshold’ in full.

Line 547: Insert ‘focus of’ so as to read: ‘… equivalent of focus of attention’

7. PLOS authors have the option to publish the peer review history of their article (what does this mean?). If published, this will include your full peer review and any attached files.

Reviewer #1: No

Reviewer #2: **Yes: **Gert-Jan Pepping

---

## [Author Response · Author response to Decision Letter 2]

2 Aug 2021

Throughout the authors should check that past tense is used when relevant. Below my pick-ups:

Line 79: Jordet suggested …

Line 121: As such, we aimed …

Line 123: the aim of this exploratory study was …

Line 243: operationalizations were made …

Line 455: initiation influenced …

Line 456: dinstance influenced …

Line 457: scans involved …

Line 458: scan revealed …

Line 564: players moved …

Line 564: until they arrived …

Line 565: they wished …

Thank you for these observations! We changed the wording to the past tense.

Lines 263/264 (and throughout): Please refer to fixation detection threshold throughout, so insert ‘fixation detection’ between ‘lower’ and ‘threshold’ here.

Line 264: It may be a while until a lower fixation detection threshold is accepted in the context of scanning. In light of this, consider replacing ‘accepted’ with ‘considered’.

These are great comments. We added fixation detection throughout and changed the wording from accepted to considered.

Lines 396-398: This sentence confused me. Maybe, for added clarity, add ‘in the video frame’ after ‘… more than seven teammates’ and after ‘… nine opponents’

Great input! We added the words as suggested.

Line 421: ‘while in defence’

Line 422/421: Consider rephrasing , eg: ‘… while in defence, no differences could be found between the amount of opponents and teammates in the video frame’.

Thank you for this suggestion. We adopted it.

Line 474: ‘the ball is being passed along the ground …’

Line 476: ‘the pass is made through the air …’

Revised as suggested.

Line 510-513: This reads like a left-over from a previous version. The reference to clear foveal information, as outline before, makes no sense from an ecological framework, and makes an implicit reference to perception of pictures. What’s more, whereas the manuscript now contributes to the readers understanding of the role of the fixation detection threshold in the determination of fixations, this sentence weakens that contributions (as you later explain in lines 516 – 523). 

Thank you for this input. We fully agree and removed the entire sentence.

Lines 652/653: As above, this sentence seems a left-over on the ‘seeing detail’ part (confusing perception and action with pictorial perception) and weakens the argument. I strongly suggest the authors rephrase or delete this very suggestive supposition that is neither based on theory nor finds any evidence in data presented.

Thank you for this input. We removed the sentence an instead added two sentences that are more in line with the story of the paper, the results, and the ecological framework of perception-action coupling: “This result can be partly explained by the adopted laboratory fixation detection threshold of 120 ms, which might be too high for unstable real-world research. However, it also implies that players, when performing scans during match-play, do not need to foveally fixate on surrounding information in order to obtain sufficient information for performing their football actions.”

Line 517/518: As above, write ‘lower fixation detection threshold’ in full.

Revised as suggested.

Line 547: Insert ‘focus of’ so as to read: ‘… equivalent of focus of attention’

Revised as suggested.

---

## [Editor Report · Decision Letter 3]

11 Aug 2021

Scanning activity of elite football players in 11 vs. 11 match play: An eye-tracking analysis on the duration and visual information of scanning

PONE-D-20-37282R3

Dear Dr. Aksum,

We’re pleased to inform you that your manuscript has been judged scientifically suitable for publication and will be formally accepted for publication once it meets all outstanding technical requirements.

Kind regards,

Greg Wood, PhD

Academic Editor

PLOS ONE
---

## [Editor Report · Acceptance letter]

13 Aug 2021

PONE-D-20-37282R3 

Scanning activity of elite football players in 11 vs. 11 match play: An eye-tracking analysis on the duration and visual information of scanning 

Dear Dr. Aksum:

I'm pleased to inform you that your manuscript has been deemed suitable for publication in PLOS ONE. Congratulations! Your manuscript is now with our production department. 

Kind regards, 

on behalf of

Dr. Greg Wood 

Academic Editor

PLOS ONE